# QuadTune version 1: A regional tuner for global atmospheric models

Vincent E. Larson[1,2], Zhun Guo[3], Benjamin A. Stephens[4], Colin Zarzycki[5], Gerhard Dikta[6], Yun Qian[2], and Shaocheng Xie[7]

[1]University of Wisconsin — Milwaukee, Milwaukee, WI, USA
[2]Pacific Northwest National Laboratory, Richland, WA, USA
[3]National Key Laboratory of Earth System Numerical Modeling and Application, Institute of Atmospheric Physics, Chinese Academy of Sciences
[4]National Center for Atmospheric Research, Boulder, CO, USA
[5]Department of Meteorology and Atmospheric Science, The Pennsylvania State University, University Park, PA, USA
[6]Department of Medical Engineering and Technomathematics, Fachhochschule Aachen, Jülich, Germany
[7]Lawrence Livermore National Laboratory, Livermore, CA, USA

**Correspondence:** Vincent E. Larson (vlarson@uwm.edu)

**Abstract.**

When a new, better-formulated physical parameterization is introduced into a global atmospheric model, aspects of the global model solutions are sometimes degraded. Then, in order to use the new global model to address science questions, there is an incentive to restore its accuracy. Oftentimes this restoration is achieved by tuning of model parameter values. Unfortunately, the retuning process is expensive because characterizing the parameter dependence requires numerous time-consuming global simulations.

To reduce the cost of tuning, this manuscript introduces a "poor man's" model tuner, "QuadTune." QuadTune carves the globe into regions and approximates the model parameter dependence through the use of an uncorrelated quadratic emulator. The simplicity of the emulator reduces the required number of global model simulations and aids explainability of tuner behavior.

Tuning removes parametric error but leaves behind model structural error. Structural error manifests itself as regional residual biases, such as stubborn biases and tuning trade-offs. To visualize these residual biases, QuadTune's software includes a set of diagnostic plots. This paper illustrates the use of the plots for characterizing residual biases with an example tuning problem.

## 1 Introduction

Global physics-based models of the atmosphere (i.e., "global models" for short) contain structural model errors. These are errors in the functional form of a model parameterization or term in the model equations. They are distinct from "parametric errors," which are errors in the values of tunable parameters in the model (Kennedy and O'Hagan, 2001; Peatier et al., 2024). Structural errors degrade the accuracy of simulated phenomena (e.g., stratocumulus clouds) that may underlie other phenomena that model users wish to study (e.g., sea-surface temperatures off the west coasts of continents). To make the model usable for

these studies, model developers frequently attempt to improve its accuracy by tuning parameter values. Unfortunately, such tuning risks building in compensating errors between model parameterizations (Mauritsen et al., 2012). In this case, there is improvement in the overall simulation but degradation in the representation of individual processes. If compensating errors exist in a global model, then later, when the model structure is improved by, say, implementing a better parameterization, the compensating error is no longer compensated, and the results may degrade. For this reason, the development of a global model often involves a two-step cycle of 1) introducing a structural improvement, and then 2) retuning parameter values to remove any degradations in accuracy due to disruption of compensating errors.

Big gains in model accuracy often come from structural model improvements (e.g., Vitart, 2014). Hence we would like to spend more time modifying the structure and less time retuning. In order to help developers quickly retune newly modified parameterizations, we seek to develop an inexpensive tuner.

Recently, a number of automated tuning methods have been developed. These include sequential methods such as Very Fast Simulated Annealing (Jackson et al., 2003, 2004; Yang et al., 2012; Zou et al., 2014). Unfortunately, performing many global atmospheric simulations in sequence incurs a long runtime. To reduce runtime, global simulations can be run in parallel by use of a perturbed parameter ensemble (PPE) (e.g., Qian et al., 2018; Li et al., 2019; Cleary et al., 2021; Dunbar et al., 2021; Hourdin et al., 2021; Eidhammer et al., 2024; Elsaesser et al., 2024). However, a PPE typically requires $O(100)$ global simulations and hence requires a large computer allocation. Sometimes a new structural modification is ready to be tried before the automated tuning is completed!

What is needed is a method to quickly re-tune global models after a structural change has been made. What is also desirable is guidance on what structural errors remain after the parametric errors have been tuned out (e.g., Rostron et al., 2025). For this purpose, the retuning need not yield the exact global optimum as long as the retuning is effective enough to indicate whether the modified parameterization has promise.

To this end, we have developed an inexpensive, "poor man's" tuner. It divides the globe into $20°$ longitude by $20°$ latitude tiles (i.e., regions) that, taken collectively, cover the globe. Our tuner treats each field within each region as a sample point. It then finds the set of parameter values of the global model that best fit the observed regional values in a least-squares sense. To do so, it uses quadratic regression (Neelin et al., 2010; Bellprat et al., 2012). Hence, we call our tuner "QuadTune." If $P$ denotes the number of tuning parameters, then QuadTune requires only $2P+1$ global simulations, because it neglects parameter interactions. Once the global runs are completed, using QuadTune to find optimal parameter values takes only seconds on a laptop computer.

QuadTune tells us those regions in which biases can be reduced and those in which they cannot. It informs the user which biases are stubborn and where there are tuning trade-offs among regions (e.g., Regayre et al., 2023; Peatier et al., 2024). In this sense, QuadTune gives hints about the nature of the model's structural error.

To visualize this information and enhance explainability (Linardatos et al., 2020), QuadTune outputs a series of diagnostic plots. These plots indicate the manner in which QuadTune has reduced regional biases and the character of the structural errors that prevent further bias reduction.

One purpose of the present paper is to document QuadTune's algorithm. Another is to illustrate the use of QuadTune's diagnostic plots. We illustrate the plots by doing an example set of tuning runs using a global model. Before we present these plots, we discuss some regression theory so that the interested reader can understand which mathematical equations are plotted in the diagnostics.

This paper is organized as follows. Section 2 describes the tuning problem that we address. Section 3 gives an outline of QuadTune's algorithm. Section 4 discusses a toy tuning problem in order to build intuition about model biases. Section 5 lists QuadTune's emulator, loss function, and some of its formulas underlying diagnostic plots. Section 6 describes the global model, parameterization, and observations used by our example tuning analysis. The example analysis is given in Section 7. Section 8 describes sensitivities to the configuration of our tuning runs. Caveats are noted in Section 9, and we conclude in Section 10.

## 2   The regional tuning problem

The mathematics of the regional tuning problem resembles that of standard least-squares (nonlinear) regression (e.g., Chapter 8 of Pollock, 1999), but there are important differences in interpretation that we broach in this section.

Suppose we have a global, physics-based atmospheric model that advances a set of fluid mechanical partial differential equations (PDEs) forward in time. Let us represent this model's set of PDEs schematically as $\mathcal{G}(t, \boldsymbol{x}; p_1, p_2)$. Here $t$ denotes time, and $\boldsymbol{x}$ denotes spatial position, e.g., latitude and longitude. In addition, $p_1$ and $p_2$ denote two physical parameters that are embedded in the model PDEs. (We limit ourselves at first to two parameters for ease of discussion.) For instance, $p_1$ and $p_2$ might denote coefficients related to turbulent dissipation. They are our input "features." We take $p_1$ and $p_2$ to be constant in time and space throughout a global simulation, but we treat them as tunable.

Although we denote the model's *set of PDEs* by $\mathcal{G}$, we denote $\mathcal{G}$'s time-averaged global-model *output* by a function, $f$:

$$f = f\left(\boldsymbol{x}; p_1, p_2\right). \tag{1}$$

Here, $f$ could represent any observable field that is output by the model. E.g., it could represent a field such as cloud cover that is part of the time-averaged model state, or it could represent a derived field, such as Shortwave Cloud Radiative Forcing (SWCF). We denote the dependence of $f$ on $\mathcal{G}$ schematically, with some abuse of notation, as

$$\overline{\mathcal{G}(t, \boldsymbol{x}; p_1, p_2)}^t \mapsto f\left(\boldsymbol{x}; p_1, p_2\right). \tag{2}$$

Here $\overline{()}^t$ denotes a time average. The particular output, $f\left(\boldsymbol{x}; p_1, p_2\right)$, that is produced depends on the particular choice of the parameter values $p_1$ and $p_2$ embedded in $\mathcal{G}$. We treat all other parameters, initial conditions, boundary conditions, etc., in $\mathcal{G}$ as known, prescribed constants.

We wish to tune $p_1$ and $p_2$ in order to improve the agreement of $\mathcal{G}$'s solutions (i.e., $f\left(\boldsymbol{x}; p_1, p_2\right)$) with observations, $f_{\text{obs}}(\boldsymbol{x})$. Here, $f_{\text{obs}}(\boldsymbol{x})$ might represent, e.g., satellite observations of cloud cover. However, no matter how $p_1$ and $p_2$ are adjusted,

$f(\boldsymbol{x}; p_1, p_2) \neq f_{\mathrm{obs}}(\boldsymbol{x})$, because $\mathcal{G}$ is not Nature's true set of PDEs and hence contains model structural error. In this initial effort, we will assume that, in comparison to $\mathcal{G}$'s structural error, the observational error in $f_{\mathrm{obs}}(\boldsymbol{x})$ is negligible. (However, Elsaesser et al. (2024) find that observational error does impact optimal parameter values. Hence observational error should be accounted for in a fuller treatment.) If $f_{\mathrm{obs}}(\boldsymbol{x})$ is taken to be an adequate representation of Nature's truth, then we may write

$$f_{\mathrm{obs}}(\boldsymbol{x}) = f(\boldsymbol{x}; p_1, p_2) + \epsilon_m(p_1, p_2), \tag{3}$$

where $\epsilon_m$ represents model structural and parametric error.

One could attempt to minimize the error at fine resolution with an integral over the whole globe:

$$\operatorname*{arg\,min}_{p_1, p_2} \int \left[ f(\boldsymbol{x}; p_1, p_2) - f_{\mathrm{obs}}(\boldsymbol{x}) \right]^2 dx. \tag{4}$$

Instead, to simplify, we carve the globe into $n$ coarse ($20° \times 20°$) tiles (i.e., regions), as, e.g., in Yarger et al. (2024). Then, assuming $f$ represents only one field, we minimize the error averaged over each region, $\boldsymbol{x}_i$:

$$\operatorname*{arg\,min}_{p_1, p_2} \sum_{i=1}^{n} \left[ \overline{f(\boldsymbol{x}; p_1, p_2)}^{\boldsymbol{x} \in \boldsymbol{x}_i} - \overline{f_{\mathrm{obs}}(\boldsymbol{x})}^{\boldsymbol{x} \in \boldsymbol{x}_i} \right]^2. \tag{5}$$

We notice two differences between a standard regression problem and our regional tuning problem. First, we assume that the residual scatter left after tuning is not due primarily to random measurement error, but is instead due mostly to model structural error in $\mathcal{G}$. Our problem is perhaps more akin to a problem of least squares function approximation, in which the observations consist of samples of a deterministic function to be matched, and $f(\boldsymbol{x}; p_1, p_2)$ is a kind of "basis function" that contains parameters to optimize (Lanczos, 1988). Second, our sample points are not drawn randomly from a population, but are instead drawn at even spatial intervals over the globe.

## 3 The QuadTune regional tuning recipe

Now that we've introduced the problem, we outline QuadTune's method. (More detail on QuadTune's method will be provided later, in Section 5.)

We define a "regional metric" — or "metric" for short — as a field (e.g., SWCF) that is output by the global model and averaged over the $i$th region. The regional metric can be compared to a reference dataset, such as an observational climatology derived from satellite measurements.

We assume that we start with observations of each regional metric. Then we perform the following steps:

1. **Preprocessing Steps:**

    (a) *Choose P model parameters to tune.* At present, QuadTune leaves the decision of which parameters ought to be tuned to expert judgment.

(b) *Choose N observed regional metrics to match.* For illustration, this paper tunes a single field, SWCF, but QuadTune allows multiple observables, e.g., SWCF and surface precipitation, to be tuned simultaneously with user-specified weighting on each observable. The choice is again left to expert judgment. To form a regional metric, the observed field must be averaged over each region, e.g., each $20° \times 20°$ tile. Here, $N$ equals the number of observed fields times the number of tiles.

(c) *Run $2P + 1$ global simulations, as follows:*

    i. Run 1 global default simulation with default parameter values.

    ii. Run $2P$ global sensitivity simulations. Perturb the parameters one at a time (Saltelli et al., 2008; Kennedy et al., 2024); that is, when one parameter is perturbed from its default value, the other parameters are kept at their default values. For example, one sensitivity simulation might be perturbed as $(p_1, p_2) \rightarrow (p_1 + \delta p_1, p_2)$. Perturb each parameter above and below its default value, by an amount decided by expert judgment, for a total of $2P$ simulations. This one-at-a-time sampling strategy determines the quadratic emulator of parameter dependence with the minimum number of global simulations.

    iii. Output the average of each regional metric to a file.

(d) *Choose regional weights, $\sigma_i$.* The user may choose to weight a region more if the user wishes to boost the chance that QuadTune will produce a good fit in that region (at the expense of other regions). In this paper, the weights are simply set proportional to the geographical area of each $20° \times 20°$ region.

2. **Tuning and Analysis Steps:**

(a) *Given the regional metrics output file and the regional weights, run QuadTune.* Upon running QuadTune, QuadTune will

    i. estimate optimal ("recommended") parameter values according to (a possibly weighted version of) Eq. (5);

    ii. estimate expected metric improvements for each region; and

    iii. generate diagnostic plots.

(b) *Optional: If desired, re-run QuadTune in order to explore tuning trade-offs.* Such experiments might delete ineffective parameters or more heavily weight a regional metric in order to see how it influences the optimal parameter values.

(c) *Run a global-model simulation with QuadTune's recommended parameter values.* Although QuadTune estimates metric improvements, the true improvements are not known until the global model is run with QuadTune's recommended parameter values.

## 4 A toy linear example of regional tuning

One of our goals is to diagnose and visualize aspects of model structural error, such as stubborn biases and tuning trade-offs among different regions. To illustrate, in a simplified setting, some of the symptoms of various kinds of structural error, we now discuss a toy example in which the emulator of parameter dependence is linear. Discussion of the quadratic term in our emulator will be deferred to Section 5 and later sections. Conceptual understanding of the toy example will aid understanding of the diagnostics that we present in Section 7 and the mathematical quantities that they plot (Appendix A).

In this example, for definiteness, we assume that $f$ denotes cloud cover. For simplicity, we tune the time-averaged cloud cover to match observations in only three regions, chosen somewhat arbitrarily. These regions are the marine stratocumulus deck off the coast of California ($Sc$), the shallow cumulus region near Hawaii ($Cu$), and the Western Pacific warm pool ($WP$). We assume that near the optimal parameter values, the model output is an approximate match to observations:

$$f(Sc; p_{1,opt}, p_{2,opt}) \approx f_{\text{obs}}(Sc)$$
$$f(Cu; p_{1,opt}, p_{2,opt}) \approx f_{\text{obs}}(Cu)$$
$$f(WP; p_{1,opt}, p_{2,opt}) \approx f_{\text{obs}}(WP). \tag{6}$$

This assumption is valid if the model structural error and observational error are not too large. Again, this has been written somewhat schematically. The "value" $\boldsymbol{x} = Sc$ in fact denotes a regional spatial average, e.g., $f_{\text{obs}}(Sc) \equiv \overline{f_{\text{obs}}(\boldsymbol{x})}^{\boldsymbol{x} \in Sc}$ and $f(Sc; p_1, p_2) \equiv \overline{f(\boldsymbol{x}; p_1, p_2)}^{\boldsymbol{x} \in Sc}$. The cloud cover in the stratocumulus region, $f(Sc)$, is an example of what this paper calls a "regional metric" or simply "metric" for short.

The functional dependence of cloud cover $f(\boldsymbol{x}; p_1, p_2)$ on $p_1$ and $p_2$ in the three regions is unknown. Ideally, we would like to map out the dependence for a broad range of values of $(p_1, p_2)$ and create a sophisticated emulator of the hills and valleys in that 2D parameter space. For instance, past authors have emulated parameter dependence by use of a Gaussian Process (e.g. Kennedy and O'Hagan, 2001; Salter et al., 2019) or a polynomial chaos expansion (Yarger et al., 2024). However, in this didactic example, we simply linearize $f(Sc; p_1, p_2)$, $f(Cu; p_1, p_2)$, and $f(WP; p_1, p_2)$ about the default values of the parameters, $(p_{1,def}, p_{2,def})$:

$$
\begin{bmatrix}
f(Sc; p_{1,opt}, p_{2,opt}) \\[2ex]
f(Cu; p_{1,opt}, p_{2,opt}) \\[2ex]
f(WP; p_{1,opt}, p_{2,opt})
\end{bmatrix}
$$

$$
\approx
\begin{bmatrix}
f(Sc; p_{1,def}, p_{2,def}) \\[2ex]
f(Cu; p_{1,def}, p_{2,def}) \\[2ex]
f(WP; p_{1,def}, p_{2,def})
\end{bmatrix}
$$

$$
+
\begin{bmatrix}
\left.\dfrac{\partial f}{\partial p_1}\right|_{\boldsymbol{x}=Sc} & \left.\dfrac{\partial f}{\partial p_2}\right|_{\boldsymbol{x}=Sc} \\[2ex]
\left.\dfrac{\partial f}{\partial p_1}\right|_{\boldsymbol{x}=Cu} & \left.\dfrac{\partial f}{\partial p_2}\right|_{\boldsymbol{x}=Cu} \\[2ex]
\left.\dfrac{\partial f}{\partial p_1}\right|_{\boldsymbol{x}=WP} & \left.\dfrac{\partial f}{\partial p_2}\right|_{\boldsymbol{x}=WP}
\end{bmatrix}
\begin{bmatrix}
\delta p_{1,opt} \\[2ex]
\delta p_{2,opt}
\end{bmatrix}
$$

$$
\approx
\begin{bmatrix}
f_{\mathrm{obs}}(Sc) \\[2ex]
f_{\mathrm{obs}}(Cu) \\[2ex]
f_{\mathrm{obs}}(WP)
\end{bmatrix}. \tag{7}
$$

Here $\delta p_{1,opt} \equiv p_{1,opt} - p_{1,def}$, and similarly for $\delta p_{2,opt}$.

What Eq. (7) assumes is that the emulator that describes the global model's parameter dependence can be approximated by linearization about the default parameter values. The linearization introduces a sensitivity matrix (or Jacobian matrix), $\mathbf{S}$, on the left-hand side. Each element of $\mathbf{S}$ represents the linear sensitivity of a particular regional metric to a particular parameter. Each row of Eq. (7) corresponds to an equation for a particular regional metric, i.e. cloud cover in the $Sc$, $Cu$, or $WP$ region.

We pause to note that errors in our problem arise from two distinct sources: 1) errors in the emulator of parameter dependence, and 2) parametric or structural errors in the global atmospheric model, $\mathcal{G}$. The latter model errors are the focus of this paper.

Now, to neaten the equation, we define the bias, $\delta b(\boldsymbol{x})$, as, e.g.,

$$
\delta b(Sc) \equiv f(Sc; p_{1,def}, p_{2,def}) - f_{\mathrm{obs}}(Sc), \tag{8}
$$

and similarly for $Cu$ and $WP$. (Note that "bias" here means a bias in model output and not a bias in a parameter value, unlike in traditional statistics nomenclature (Ross, 2009).) Then we may move the default values to the right-hand side of Eq. (7):

$$
\begin{bmatrix}
\left.\frac{\partial f}{\partial p_1}\right|_{\boldsymbol{x}=Sc} & \left.\frac{\partial f}{\partial p_2}\right|_{\boldsymbol{x}=Sc} \\[2ex]
\left.\frac{\partial f}{\partial p_1}\right|_{\boldsymbol{x}=Cu} & \left.\frac{\partial f}{\partial p_2}\right|_{\boldsymbol{x}=Cu} \\[2ex]
\left.\frac{\partial f}{\partial p_1}\right|_{\boldsymbol{x}=WP} & \left.\frac{\partial f}{\partial p_2}\right|_{\boldsymbol{x}=WP}
\end{bmatrix}
\begin{bmatrix}
\delta p_{1,opt} \\[2ex]
\delta p_{2,opt}
\end{bmatrix}
\approx -
\begin{bmatrix}
\delta b(Sc) \\[2ex]
\delta b(Cu) \\[2ex]
\delta b(WP)
\end{bmatrix},
\tag{9}
$$

or, rewritten in symbolic form,

$$
\mathbf{S} \cdot \delta \boldsymbol{p}_{opt} \approx -\delta \boldsymbol{b}.
\tag{10}
$$

We regard as knowns the bias vector $\delta \boldsymbol{b}$ and the sensitivity matrix $\mathbf{S}$.

The matrix equation (9) has no exact solution, in general, because it has more equations (rows) than unknowns (columns). However, one can find the optimal parameter values in a least-squares sense by means of linear regression (Press et al., 2007, Section 15.4). In this analogy, $\mathbf{S}$ corresponds to a design matrix in linear regression, and each row of $\mathbf{S}$ — i.e., each regional metric — may be interpreted as a "sample point" drawn from a distribution of sensitivities. The "sample point" is a multivariate sample of the sensitivities of $f(\boldsymbol{x}; p_1, p_2)$ to the parameters $p_1$ and $p_2$ in a region $\boldsymbol{x}_i$.

In our problem, which neglects observational error, the scatter about the regression curve is a consequence of deterministic structural errors in the global model, $\mathcal{G}$. Consequently, the scatter provides clues about where the errors in $\mathcal{G}$ lie, and hence the scatter is a primary object of interest.

## 4.1 The column-space geometric interpretation of the sensitivity matrix

How does model structural error manifest itself in the context of parametric tuning? We can gain some basic understanding, for linear parameter dependence, by interpreting the sensitivity matrix, $\mathbf{S}$, as a set of column vectors. From the column-vector point of view, the goal of tuning is to represent, insofar as possible, the bias column vector $\delta \boldsymbol{b}$ as a linear combination of the column vectors of $\mathbf{S}$ (see Eq. 9 above and Chapter 8 of Pollock (1999)). Suppose that $\delta \boldsymbol{b}$ has length $N$ (i.e., $N$ regional metrics) and that there are $P$ column vectors of $\mathbf{S}$ (i.e., $P$ parameters). In the usual circumstance, $N > P$, there are more equations than unknowns, and hence exact representation of $\delta \boldsymbol{b}$ is impossible in general. Instead, solving the least-squares optimization problem (5) leaves a residual bias. The residual bias is a consequence of the fact that $\mathcal{G}$ has a model structural error. In contrast, in the special case that $\delta \boldsymbol{b}$ happens to reside within the $P$-dimensional subspace that is spanned by the columns of $\mathbf{S}$, then $\delta \boldsymbol{b}$ can indeed be exactly represented as a linear combination of them. If so, then the bias can be removed entirely by changes in parameter values, indicating that the global model, $\mathcal{G}$, has no structural error.

The $j$th column of $\mathbf{S}$ tells us how perturbing the $j$th parameter, $\delta p_j$, affects the spatial pattern of a metric across different regions of the globe. At first one might think that if all elements of the $j$th column of $\mathbf{S}$ are small, relative to those of other

columns, then all regional metrics are relatively insensitive to the corresponding parameter, $\delta p_j$. A naive sensitivity analysis might then drop $\delta p_j$ from the set of tuning parameters. However, in this linear problem, this lack of sensitivity could, in principle, be counteracted simply by increasing the magnitude of the parameter perturbation $\delta p_j$. Rather, the true problem occurs when some elements of the column are small and others are large, in such a way that increasing $\delta p_j$ to remove the bias in an insensitive metric (i.e., row) creates a large error in a more sensitive metric (i.e., row). Then adjustment of $\delta p_j$ is unable to remove the bias in the insensitive region. In practice, the rows of $\mathbf{S}$ are unlikely to include all sensitive regional metrics that could be observed, and hence an unduly large increase in $\delta p_j$ risks incurring a large error in an excluded metric.

Model structural error can be quantified by the residual bias that remains after tuning out the parametric error. Assume, for simplicity, that $\mathbf{S}$ has no zero singular values (Press et al., 2007, Section 15.4). Now suppose that we use least-squares linear regression to find optimal parameter perturbations $\delta p_{opt,j}$. Then let us define

$$\delta b_{remov,i} \equiv -S_{ij}\delta p_{opt,j}, \tag{11}$$

where $S_{ij}$ is the $ij$th element of the matrix $\mathbf{S}$. Also, $\delta \boldsymbol{b}_{remov}$ is the default model output minus the tuned model output. The vector $\delta \boldsymbol{b}_{remov}$ is the part of the bias $\delta \boldsymbol{b}$ that is removable by linear regression. Thinking more geometrically, $\delta \boldsymbol{b}_{remov}$ is the part of the bias vector $\delta \boldsymbol{b}$ that lies within the subspace spanned by the columns of $\mathbf{S}$.

Now define the residual bias, $\delta \boldsymbol{b}_{resid}$, as the part of the bias that remains after $\delta \boldsymbol{b}_{remov}$ has been removed by linear regression:

$$\delta \boldsymbol{b} \equiv \delta \boldsymbol{b}_{remov} + \delta \boldsymbol{b}_{resid}. \tag{12}$$

Here, $\delta \boldsymbol{b}_{resid}$ is the tuned model output minus the observational values. (Note that the residual bias $\delta \boldsymbol{b}_{resid}$ is defined to have the opposite sign as the residual that is traditionally defined in statistics (Pollock, 1999).) $\delta \boldsymbol{b}_{resid}$ lies outside the subspace spanned by the columns of $\mathbf{S}$. (In fact, in linear regression, $\delta \boldsymbol{b}_{resid}$ turns out to be orthogonal to $\delta \boldsymbol{b}_{remov}$ (Pollock, 1999).)

Hence, $\delta \boldsymbol{b}_{resid}$, supplemented with $\delta \boldsymbol{b}$, provides information about structural errors in $\mathcal{G}$ such as stubborn biases and tuning trade-offs (e.g., Peatier et al., 2024). Namely, we define the bias in the $i$th metric to be stubborn if

$$|\delta b_{remov,i}| << |\delta b_i|$$
$$\text{so that} \quad |\delta b_{resid,i}| \sim |\delta b_i|. \tag{13}$$

Also, we define a tuning trade-off between metrics $l$ and $i$ to be the situation in which tuning improves region $l$ at the expense of region $i$ (or vice-versa):

$$|\delta b_{resid,l}| < |\delta b_l| \qquad \text{improved region}$$
$$\text{and} \quad |\delta b_{resid,i}| > |\delta b_i| \qquad \text{traded-off region.} \tag{14}$$

## 4.2 The row-space geometric interpretation of the sensitivity matrix

The previous section examined the vector space of $\mathbf{S}$'s columns. In this section, we examine the vector space of $\mathbf{S}$'s rows. The elements of the $i$th row of $\mathbf{S}$ contain the sensitivities of the regional metric $i$ to each of the $P$ parameters.

From the perspective of the row-vector space, the goal of (linear) tuning is to find a single parameter perturbation vector, $\delta \boldsymbol{p}$, whose dot product with the $i$th row of $\mathbf{S}$ matches $-\delta b_i$ as closely as possible, for all $i$ (see Eq. 10). However, this goal commonly encounters two difficulties, namely, stubborn biases and tuning trade-offs.

A stubborn bias occurs when all elements (sensitivities) of the $i$th row of $\mathbf{S}$ are small ($\epsilon$), but the bias $\delta b_i$ is large in magnitude. This can be illustrated by the following example matrix equation snippet:

$$
\begin{bmatrix} \cdots & \cdots \\ \epsilon & \epsilon \\ 1 & 2 \\ \cdots & \cdots \end{bmatrix}
\begin{bmatrix} \delta p_{1,opt} \\ \delta p_{2,opt} \end{bmatrix}
\approx -
\begin{bmatrix} \cdots & \cdots \\ 1 \\ 2 \\ \cdots & \cdots \end{bmatrix} .
\tag{15}
$$

In this case, it is impossible to remove the bias in the $i$th row with sensitivities $\epsilon$ without choosing a large-magnitude parameter perturbation, $|\delta \boldsymbol{p}|$. Choosing large $|\delta \boldsymbol{p}|$ is problematic, in our experience, for at least two reasons. First, even if large $|\delta \boldsymbol{p}|$ reduces the bias in the $i$th regional metric, it is likely to cause degrading side effects in other, more sensitive regional metrics, such as row $i+1$. (This problem was discussed in Section 4.1 from the column-space point of view.) Second, if any component of $\delta \boldsymbol{p}$ strays outside the low or high values of the sensitivity runs, then it is more likely there will be a violation of the assumption of QuadTune's emulator that $\mathcal{G}$ can be represented by a simple quadratic interpolation. Consequently, when $|\delta \boldsymbol{p}|$ is large, the parameter-value recommendations of QuadTune's emulator may lie far from the optimum of the actual global model.

The question of whether a regional bias is stubborn is different from the question of whether a parameter is non-influential. A typical sensitivity analysis, as practiced in the atmospheric sciences, determines whether a particular *parameter* has little *influence* over most of the globe (Saltelli et al., 2008). This involves comparing one column of the sensitivity matrix, $\mathbf{S}$, to another. Analyzing a stubborn bias asks a different question: Is a particular *regional metric* insensitive to *all* parameters? Deducing whether the $i$th regional metric is unbudgeable involves comparing the magnitude of the $i$th row of $\mathbf{S}$ and the corresponding bias $\delta b_i$ (i.e., the $i$th metric) with that of another row (i.e., another metric).

Another tuning problem is tuning trade-offs (Neelin et al., 2010; Peatier et al., 2024), as defined in Eq. (14). For example, if two rows of $\mathbf{S}$ are proportional to each other, then adjusting the parameters has a proportional effect on, e.g., clouds in the two corresponding regions. Then we do not have the ability to brighten the clouds independently in the two regions. This poses a

difficulty if the biases are different in the two regions, as in the following snippet of a matrix equation:

$$
\begin{bmatrix}
\cdots & \cdots \\
2 & 1 \\
4 & 2 \\
\cdots & \cdots
\end{bmatrix}
\begin{bmatrix}
\delta p_{1,opt} \\
\delta p_{2,opt}
\end{bmatrix}
\approx -
\begin{bmatrix}
\cdots & \cdots \\
-3 \\
6 \\
\cdots & \cdots
\end{bmatrix} . \tag{16}
$$

Then there will be a tuning trade-off. This might happen, e.g., if the parameters brighten stratocumulus and cumulus regions similarly, and in the default simulation, the stratocumulus is too dim while the cumulus is too bright. Note that the existence of
255 proportional rows in $\mathbf{S}$ is not necessarily problematic if the right-hand side biases associated with the two rows are consistent. Also note that nearly but not exactly proportional rows will, in principle, allow both regions to be fit but only at the cost of increasing the magnitude of the parameter values.

If there are tuning trade-offs among different regional metrics, tuning may leave the parameter values little changed, even if a set of parameters is deemed sensitive, and a set of metrics is deemed budgeable. Therefore a standard parameter sensitivity
analysis (e.g., Nardi et al., 2022, 2024) is helpful but insufficient; tuning is necessary.

To overcome either stubborn biases or tuning trade-offs, a developer must either find a new parameter, or else the developer must make a model structural change.

## 5 QuadTune's emulator, loss function, and quasi-linear approximations

Section 4 discussed a toy emulator of parameter dependence that is linear because it provides a simple didactic illustration of
265 various impediments to tuning away errors. In fact, however, QuadTune's emulator extends beyond the linear terms to include the diagonal part of the quadratic terms (Neelin et al., 2010; Bellprat et al., 2012). Including these extra terms has the drawback of requiring an extra $P$ global simulations beyond the $P+1$ simulations needed for a linear emulator, leading to a total of $2P+1$ simulations. On the other hand, retaining the diagonal quadratic terms has two advantages. First, the quadratic terms help regularize the optimized parameter values. That is, they help limit the size of the parameter perturbations during tuning.
This is helpful because, in our experience, large perturbations tend to have damaging side effects (see Section 4.2). Second, including the quadratic terms better approximates the global model's parameter dependence, which can be strongly nonlinear.

In this section, we will discuss QuadTune's quadratic emulator, QuadTune's loss function, and some approximate quasi-linear functions that we visualize in our diagnostics.

### 5.1 QuadTune's quadratic, non-interacting emulator

We wish to build an emulator for the model's parameter dependence. The $i$th of $N$ simulated regional metrics, $m_i$, is given by

$$m_i \equiv m_i(\boldsymbol{p}) \equiv \overline{f(\boldsymbol{x};\boldsymbol{p})}^{\boldsymbol{x}_i}. \tag{17}$$

Now we normalize so that tuning will fairly consider all parameters and metrics. In order to make the magnitudes of the parameters similar to each other, we normalize them by the default-simulation value of $p_j$, $p_{j,def}$.

$$\tilde{p}_j = \frac{p_j}{|p_{j,def}|}. \tag{18}$$

Likewise, we also normalize the simulated metrics and biases by the globally averaged observed values of the metrics, $m_{\mathrm{obs}}$:

$$\tilde{m}_i = \frac{m_i}{|m_{\mathrm{obs}}|} \tag{19}$$

and

$$\delta\tilde{b}_i = \frac{\delta b_i}{|m_{\mathrm{obs}}|}. \tag{20}$$

These normalizations are a kind of weighting. Other normalizations are plausible and will yield different answers. Henceforth,
having normalized the equations, we will use normalized variables and drop the tildes in the following sections and appendices.

We approximate the model's parameter dependence, i.e., the functions $m_i(\boldsymbol{p})$, using a quadratic expansion. This simple expansion prevents QuadTune from doing more than seeking a nearby local minimum in parameter space. However, the simplicity of the approximation helps us better understand structural errors.

In addition to the linear terms in the polynomial expansion, we also include the quadratic terms in order to better represent
the parameter dependence about the default value, $\boldsymbol{p}_{def}$:

$$m_i(\boldsymbol{p}) = m_i(\boldsymbol{p}_{def}) + \sum_{j=1}^{P} \frac{\partial m_i}{\partial p_j} \delta p_j + \frac{1}{2} \sum_{j=1}^{P} \sum_{k=1}^{P} \delta p_k \frac{\partial^2 m_i}{\partial p_k \partial p_j} \delta p_j + .... \tag{21}$$

Here we recognize that the linear term is simply the sensitivity matrix $S_{ij} \equiv \partial m_i/\partial p_j$.

The core of the quadratic term is a 3D tensor, $\frac{\partial^2 m_i}{\partial p_k \partial p_j}$, that includes both diagonal terms ($k = j$) and cross terms or interaction terms ($k \neq j$). Retaining the interaction terms would improve the emulator, but in the most straightforward implementation
would also require a large number of extra global simulations. To avoid this expense, we drop the interaction terms. Doing so diminishes the accuracy of the emulator, of course. However, even when parameter perturbations extend to the limits recommended by expert judgment, parameter interactions have been found to be relatively small in both global and single-column atmospheric simulations. For instance, in the global PPE of Qian et al. (2018), parameter two-way interactions have a relative contribution of 5 to 10% (see their Fig. 2). Similarly, the global simulations of Neelin et al. (2010) obtain similar optimal

parameter values when the interaction terms are kept or omitted. In the single-column stratocumulus simulations of Guo et al. (2014), the relative contribution of parameter two-way interactions is usually less than 5% (see their Fig. 3).

Once we drop the interaction terms, then QuadTune's emulator becomes

$$m_i(\boldsymbol{p}) = m_i(\boldsymbol{p}_{def}) + \sum_{j=1}^{P} \frac{\partial m_i}{\partial p_j} \delta p_j + \frac{1}{2} \sum_{j=1}^{P} \frac{\partial^2 m_i}{\partial p_j^2} (\delta p_j)^2 + \epsilon_{e,i}. \tag{22}$$

Here, $\epsilon_{e,i}$ represents error in the $i$th regional metric of the emulator (i.e., error in the omission of high-order polynomial terms). With no interaction terms, the quadratic emulator function becomes a simple uncorrelated parabola. The diagonal quadratic term in QuadTune's loss function (22) is reminiscent of the shrinkage penalty term that helps regularize ridge regression (James et al., 2013).

## 5.2 QuadTune's loss function

We define error in the $i$th regional metric produced by the global model, $\epsilon_{m,i}$ (which is distinct from error in the emulator, $\epsilon_{e,i}$) by

$$f_{\text{obs},i} \equiv \overline{f_{\text{obs}}(\boldsymbol{x})}^{\boldsymbol{x}_i} = m_i(\boldsymbol{p}) + \epsilon_{m,i}(\boldsymbol{p}). \tag{23}$$

We define the bias by

$$\delta b_i \equiv m_i(\boldsymbol{p}_{def}) - f_{\text{obs},i}. \tag{24}$$

Then the approximated equation becomes

$$-\delta b_i = \sum_{j=1}^{P} \frac{\partial m_i}{\partial p_j} \delta p_j + \frac{1}{2} \sum_{j=1}^{P} \frac{\partial^2 m_i}{\partial p_j^2} (\delta p_j)^2 + \epsilon_i. \tag{25}$$

where we have combined the errors in the emulator and the global model: $\epsilon_i = \epsilon_{e,i} + \epsilon_{m,i}$.

If we want to manually alter the influence of the $i$th regional metric, then we may multiply the entire $i$th equation, including the non-linear term, by a weight, $\sigma_i$:

$$-\sigma_i \delta b_i = \sigma_i \sum_{j=1}^{P} \frac{\partial m_i}{\partial p_j} \delta p_j + \frac{1}{2} \sigma_i \sum_{j=1}^{P} \frac{\partial^2 m_i}{\partial p_j^2} (\delta p_j)^2$$
$$+ \sigma_i \epsilon_i + \dots$$
$$(\text{no sum over } i). \tag{26}$$

In this paper, the weights are simply taken to be proportional to the geographic areas of the $20° \times 20°$ regions.

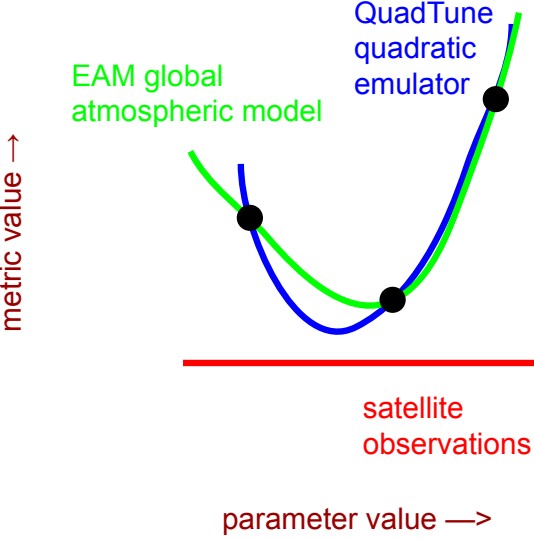

**Figure 1.** Schematic diagram illustrating the parameter dependence of a quadratic emulator (blue), a global atmospheric model (green), and observations (red). The emulator is constructed to match the global model at three parameter values (black dots). We are ultimately interested in finding the parameter value that minimizes the distance between the global model and the observations, but for efficiency we in fact find the parameter value that minimizes the distance between the emulator and the observations. The error in emulator shape is less consequential if the global model varies smoothly with a change in parameter value and if the optimized value stays within the parameter range spanned by the three parameter values.

To find the optimal parameter values, QuadTune minimizes the following least-squares loss function, $L$:

$$L \equiv \sum_{i=1}^{N} \sigma_i^2 \left[ -\delta b_i - \sum_{j=1}^{P} \left( \frac{\partial m_i}{\partial p_j} \delta p_j + \frac{1}{2} \frac{\partial^2 m_i}{\partial p_j^2} (\delta p_j)^2 \right) \right]^2 . \tag{27}$$

Although QuadTune's emulator is quadratic, its loss function is quartic. Because the loss function (27) is quartic, it is not necessarily convex and therefore does not necessarily have a single unique minimum.

To do this minimization, QuadTune calls SciPy's `optimize.minimize` package with the Powell method, which performs a sequence of one-dimensional minimizations along conjugate directions in parameter space (Powell, 1964; Press et al., 2007). For a quadratic function, Powell's method requires $P(P+1)$ 1D minimizations. It scales linearly with the number of regional metrics.

As input to the minimization routine, we must provide $\partial m_i/\partial p_j$ and $\partial^2 m_i/\partial p_j^2$ for each $i$ and $j$. These are calculated by fitting a parabola in $p_j$ to $m_i(p_j)$. The parabola passes through the 3 points formed by the output of the default simulation and the outputs of the 2 simulations that perturb $p_j$ (Fig. 1). Therefore, at these three points, QuadTune's parabolic emulator is an exact match to the global model solutions. Despite the resemblance of the quadratic emulator (22) to a Taylor series, (22)

is, strictly speaking, a polynomial interpolation, rather than a Taylor series. That is, the derivatives in (22) — $\partial m_i / \partial p_j$ and $\partial^2 m_i / \partial p_j^2$ — are not guaranteed to match the global model's derivatives at the default parameter value.

From the coefficients of the parabola, the first and second derivatives of the parabola are calculated at the default value of $p_j$. QuadTune's one-at-a-time sampling strategy, which perturbs each parameter high and low, is designed to estimate $\partial m_i / \partial p_j$ and $\partial^2 m_i / \partial p_j^2$ with the minimum number of global simulations.

### 5.3 Quasi-linear approximations that are useful for diagnostic plots

We now write down some formulas that are not needed for finding optimal parameter values, but nevertheless are useful for
creating visual diagnostics. In particular, we have found that QuadTune's behavior is not well described by the linear sensitivity matrix, $\mathbf{S}$. Therefore, we construct an extended matrix, $\mathbf{S}^+$, that captures some non-linearity.

After QuadTune has found the optimal parameter perturbation $\delta \boldsymbol{p}_{opt}$, we can linearize the quadratic term about it ex post facto. This yields an approximate quasi-linear form of (26):

$$
-\sigma_i \delta b_i \approx \sigma_i \sum_{j=1}^{P} \left( \frac{\partial m_i}{\partial p_j} + \frac{1}{2} \delta p_{j,opt} \frac{\partial^2 m_i}{\partial p_j^2} \right) \delta p_j
$$
$$
+ \sigma_i \epsilon_i + ...
$$
$$
\text{(no sum over } i). \tag{28}
$$

In order to generalize the linear sensitivity matrix to a quasi-linear form, we define the matrix

$$
S_{ij}^+ (\delta \boldsymbol{p}_{opt}) \equiv \frac{\partial m_i}{\partial p_j} + \frac{1}{2} \delta p_{j,opt} \frac{\partial^2 m_i}{\partial p_j^2}
$$
$$
\text{(no sum over } j). \tag{29}
$$

If we substitute $S_{ij}^+$ into (28), then the quadratic regression formula can be written as a simple quasi-linear matrix multiplication:

$$
-\sigma_i \delta b_i \approx \sigma_i \sum_{j=1}^{P} S_{ij}^+ (\delta \boldsymbol{p}_{opt}) \delta p_j + \sigma_i \epsilon_i + ...
$$
$$
\text{(no sum over } i). \tag{30}
$$

Equation 30 indicates that the $i$th bias is approximated by summing the contributions from each of the $P$ parameter perturbations $\delta p_j$. However, for diagnostic purposes, it is sometimes helpful to keep the parameter contributions separate from each other. To do so, we define a new matrix

$$
T_{ij}^+ \equiv S_{ij}^+ (\delta \boldsymbol{p}_{opt}) \delta p_{j,opt} \quad \text{(no sum over } j). \tag{31}
$$

From (30) and (31), we see that the bias for the $i$th regional metric, $\delta b_i$, is approximated by a sum over the $i$th row of $T_{ij}^+$. The
355 $ij$th element of $T_{ij}^+$ represents the total contribution to the $i$th regional metric of tuning the $j$th parameter. It takes into account both the sensitivity $S_{ij}^+$ and the size of the optimal parameter perturbation, $\delta p_{j,opt}$.

## 6 The turbulence and cloud parameterization, global atmospheric model, and SWCF observations that we use

Here we overview the turbulence and cloud parameterization (Cloud Layers Unified By Binormals, CLUBB, Larson (2017)) whose parameters we tune. We also overview the global model that hosts CLUBB, namely, the Energy Exascale Earth System Model (E3SM, Golaz et al. (2022)), or more precisely, its atmosphere component (EAM, Rasch et al. (2019)). Finally, we note the observations that we attempt to match in our example tuning analysis.

### 6.1 CLUBB model description

CLUBB is a parameterization of subgrid-scale clouds and turbulence in atmospheric models (Larson, 2017). Given mean profiles of winds, moisture, and temperature, CLUBB estimates vertical turbulent fluxes of those fields and also estimates subgrid cloud fraction and liquid water content.

CLUBB prognoses various turbulence moments, and its prognostic equations contain damping time scales related to pressure damping and turbulent dissipation. The version of CLUBB in the default version of EAM uses the so-called "Lscale" code option (Golaz et al., 2002). In simple overview, the CLUBB-Lscale option approximates the turbulent mixing length scale as the distance that a test parcel can move up or down before reaching its level of neutral buoyancy. In contrast, the version of CLUBB retuned here parameterizes the damping time scales by use of the so-called "taus" code option. CLUBB-taus uses simple diagnostic formulas to estimate the effects of physical processes such as vertical wind shear and buoyant stratification. The most relevant aspects of CLUBB-taus are listed in Appendix B, but for more details, see Guo et al. (2021) and Zhang et al. (2023). In order to switch from CLUBB-Lscale to CLUBB-taus, one sets the CLUBB flag `l_diag_Lscale_from_tau = .true.`. Doing so, unfortunately, leads to model biases that we attempt to tune away. This paper uses a version of CLUBB from 28 Nov 2022: https://github.com/larson-group/clubb_release/commit/c50ab36d29f7c3cdbadbfa6cfa5cf935451e26a2 .

The CLUBB parameters that we tune are described in Table B1. Since we do not tune non-CLUBB parameters, we do not sample the global model's full parametric error. However, our tuning run is intended to be merely an example demonstration. QuadTune is capable of tuning parameters in other model components if so desired.

### 6.2 Global atmospheric model

EAM is a global atmospheric model that calls CLUBB in order to estimate the effects of small-scale clouds and turbulence. The global-model code base we use is a development version of EAM that is a close predecessor to EAMv3.0.0 (Xie et al., 2025). In particular, the development version includes major new features of EAMv3, such as the Predicted Particle Properties (P3) stratiform microphysics scheme, convective microphysics, a mass-flux adjustment for the Zhang-McFarlane deep convective scheme, and the Multiscale Coherent Structure Parameterization (MCSP) (Terai et al., 2024). Use of a development version of EAM is appropriate to our goal, which is to illustrate the use of QuadTune for retuning after making a structural modification, in our case, from CLUBB-Lscale to CLUBB-taus.

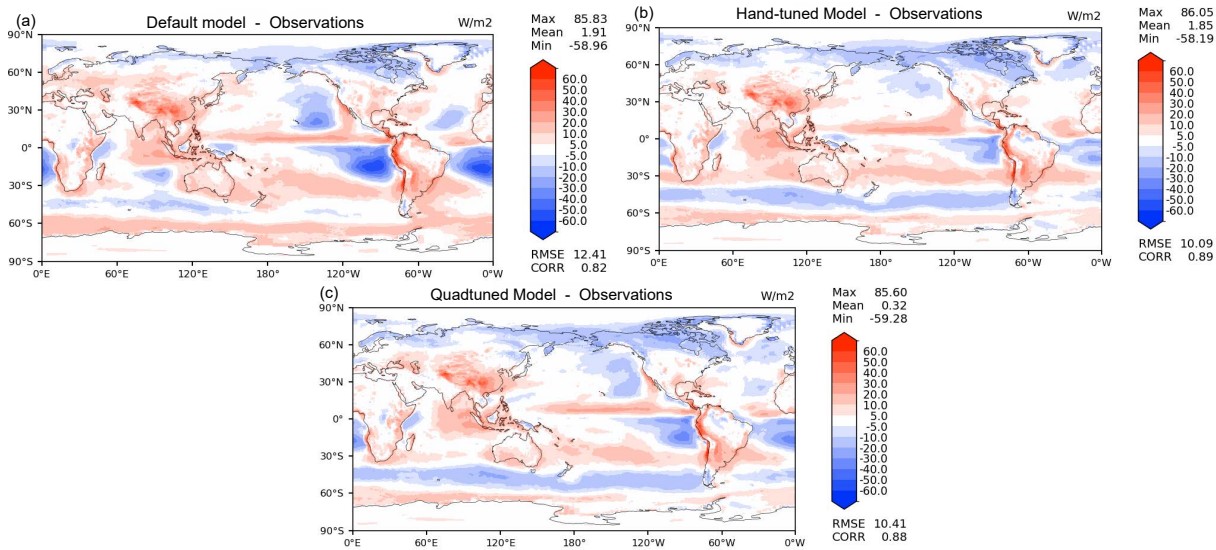

**Figure 2.** Biases of SWCF in various versions of EAM-taus: (a) default (untuned), (b) hand-tuned, and (c) quad-tuned. Without tuning, the stratocumuli in EAM-taus are too bright. These biases can be reduced almost as much by automated quad-tuning (c) as they can by laborious hand-tuning (b).

To produce the results in this paper, we branched off of branch https://github.com/E3SM-Project/v3atm/commits/NGD_v3atm/ at commit https://github.com/E3SM-Project/v3atm/commit/555c7b81080c1e5262f1ec56052787819d984ad8, which was made on 22 Jan 2023. (EAMv3.0.0 was released on 4 Mar 2024.)

### 6.3 Observations

The main goals of this paper are to document QuadTune's algorithm and to illustrate how to use its diagnostic plots. For these limited purposes, it is sufficient to tune to observations of a single variable, namely Shortwave Cloud Radiative forcing (SWCF). SWCF measures the radiative perturbation due to the presence of clouds. The more negative SWCF, the brighter the cloud, because brighter clouds reflect more incoming shortwave radiation, thereby reducing the net shortwave flux at the top of the atmosphere.

The observational dataset of SWCF that we use is version 4.1 of Clouds and the Earth's Radiant Energy System (CERES) Energy Balanced And Filled (EBAF) (Loeb et al., 2018).

## 7 An example tuning analysis: Matching observations of SWCF

QuadTune not only finds optimal values of tuning parameters but also generates diagnostic plots. These plots are designed to characterize regional biases in the global model solutions and the dependence of those solutions on parameter values. For instance, the plots indicate 1) the relative importance of nonlinear parameter dependencies versus linear parameter sensitivities;

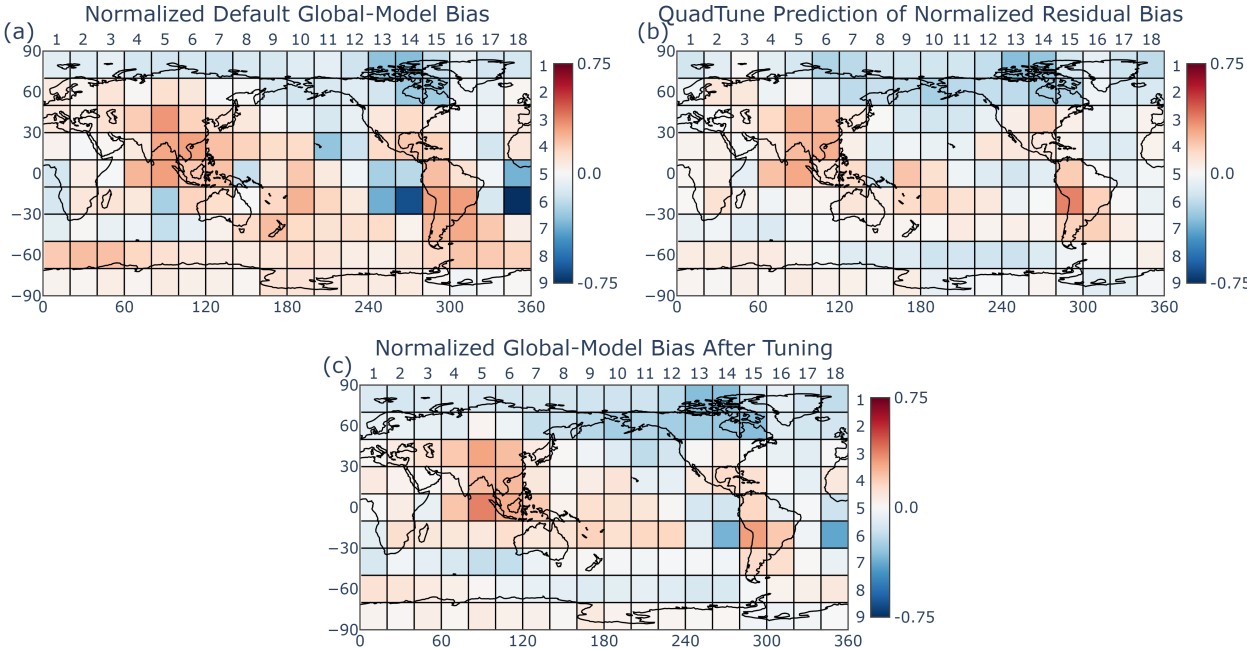

**Figure 3.** Global output of normalized SWCF averaged over $20° \times 20°$ regions. Panel a) shows the normalized model biases (model minus observations) from the default EAM simulation. Panel b) shows QuadTune's prediction of residual biases after tuning. Panel c) shows the actual residual biases of a EAM simulation that uses QuadTune's recommended parameter values. Before tuning, stratocumulus clouds (regions 6_14 and 6_18) are much too bright (a). QuadTune thinks that it can diminish those biases (b), but only at the cost of worsening biases elsewhere, e.g., over Eastern United States (region 3_14) and over the Southern Hemisphere storm track (region 8_13). An EAM simulation using QuadTune's recommended parameter values (c) shows reduced stratocumulus biases, but not as much as QuadTune predicts.

2) which regional biases can be removed by tuning; 3) which parameters are most helpful in removing those biases; and 4) the nature of the biases that cannot be removed. The mathematical quantities plotted are listed in Appendix A.

To illustrate the use of these diagnostic plots, we now tune the SWCF field produced by the updated taus version of EAM described in Section 6.2. We tune $P = 5$ tunable parameters with the names c8, n2_thresh, sfc, n2, and n2_wp2. They are defined in Table B1.

### 7.1 How much bias can be removed by a quadratic emulator without parameter interactions?

The default version of EAM-taus is created by transplanting CLUBB-taus, which was described in Section 6.1, into a version of EAM that has been tuned around CLUBB-Lscale. Consequently, EAM-taus has an unimpressive RMSE of SWCF of 12.4 W m$^{-2}$, in part because the simulated stratocumulus (Sc) clouds off the coasts of Chile (South America) and Namibia (Africa) are far too bright (see Fig. 2(a), blue contours). From this starting point, in which the model is badly out of tune, much of the parametric error can be removed by QuadTune, despite the simplicity of its emulator.

Hand tuning of EAM-taus dims the Sc and improves the RMSE of SWCF to 10.1 W m$^{-2}$ (Fig. 2(b)). However, hand tuning can require dozens of simulations and months of work. To avoid the labor of hand tuning, we instead employ QuadTune. That is, we tune EAM-taus using QuadTune, we set the parameter values in EAM-taus to the values recommended by QuadTune, and then we re-run EAM-taus with the recommended parameter values. This yields a SWCF RMSE of 10.4 W m$^{-2}$ (Fig. 2(c)). Although QuadTune's RMSE (10.4 W m$^{-2}$) is worse than the hand-tuned RMSE (10.1 W m$^{-2}$), QuadTune's global-mean bias (0.32 W m$^{-2}$) is better than the hand-tuned bias (1.85 W m$^{-2}$). Furthermore, QuadTune makes a sizable improvement over the untuned, default RMSE (12.4 W m$^{-2}$). However, we add the caveat that an equally sizable improvement would not be expected if the default model were already highly tuned to begin with.

QuadTune not only recommends parameter values but also estimates how much of the regional biases in the default simulation will be removed if EAM is run with QuadTune's recommended parameter values. In our example tuning run, QuadTune thinks that the use of its recommended parameter values can reduce the EAM-taus Sc biases almost to zero (see boxes 6_14 and 6_18 in Fig. 3(b)). However, in this case QuadTune is too optimistic; in fact, when its recommended values are used, the stratocumulus bias is not even halved (Fig. 3(c)). QuadTune's prediction is imperfect because its simple quadratic emulator is approximate. Nevertheless, QuadTune's prediction of the bias reduction is qualitatively correct and useful.

## 7.2 The importance of nonlinear parameter dependence

The parameter dependence is relatively easy to intuit when the first-order derivatives, $\partial m_i / \partial p_j$ ("sensitivities") are more important than the second-order derivatives, $\partial m_i^2 / \partial p_j^2$ ("curvature terms" or Hessian terms, Section 15.5 of Press et al. (2007)). Then, with regard to a given metric $m_i$ and a given parameter $p_j$, it is sensible to speak of a *single* sensitivity. In contrast, when the curvature terms matter, then the sensitivity *varies* over the relevant region of parameter space. As a consequence, hard-won intuition about how a parameter influences model output for a default set of parameter values may mislead us when we consider a model configuration with a different default set of parameter values.

In our tuning example, non-linear parameter dependence does matter (see the "three-dot" plot, Fig. 4). While linear parameter dependence dominates the sensitivities of some regions (e.g. the stratocumulus regions 6_14 and 6_18), nonlinear parameter dependence strongly influences other regions (e.g. 3_6 in China or 8_13 in the SH storm track). These nonlinear sensitivities are not isolated examples. Considering all regions over the globe, the quadratic contributions to sensitivity are smaller than the linear contributions but still sizable (not shown). Nonlinear parameter dependence was also found to be important in the global model of Elsaesser et al. (2024).

In principle, one could attempt to relate these strong nonlinear dependencies to how these parameters enter EAM's equations. However, doing so would be a difficult task that is beyond the scope of this manuscript. Here, we merely speculate on the reason for one simple example of nonlinearity, namely, the fact that c8 (see Table B1) sometimes has weaker sensitivity for smaller values of c8 (see the left-most column of Fig. 4). The reason possibly has to do with the fact that c8 is the coefficient of a damping term in the $\overline{w'^3}$ equation that competes with another damping term, the $C_{11}$ damping term, in the same equation (see Eq. B2). When c8 is small, the $C_{11}$ term presumably dominates, rendering the value of c8 less significant.

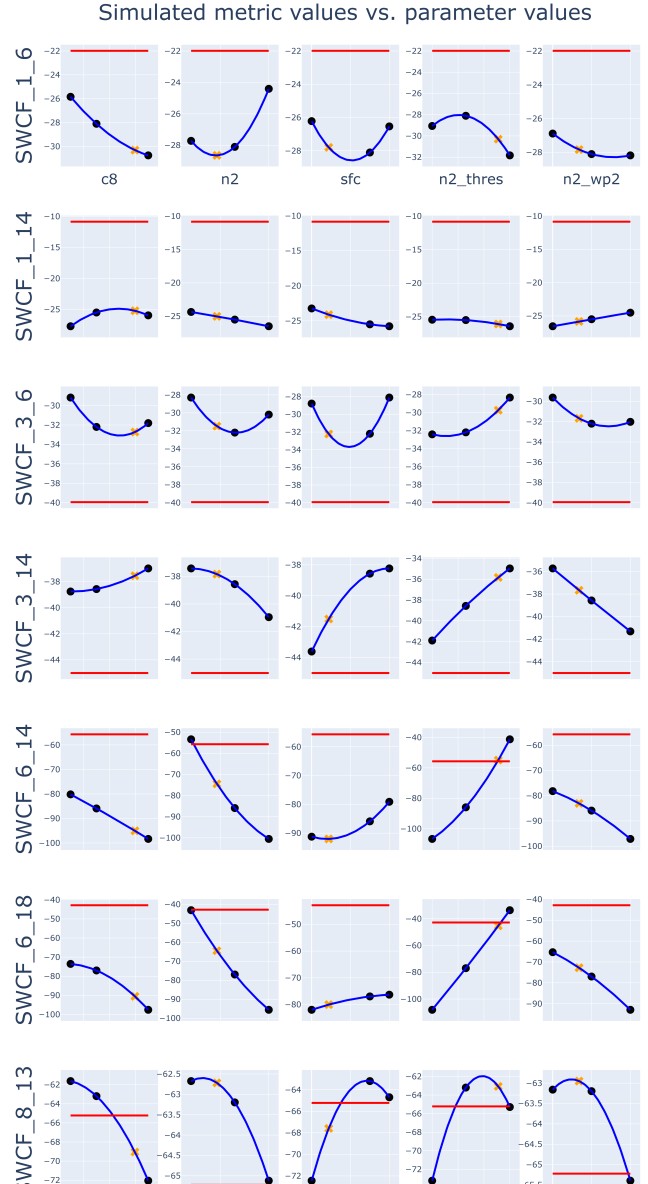

**Figure 4.** "Three-dot" plot showing the SWCF values from the sensitivity and default runs (three black dots, with the middle dot being the default), quad-tuned tuning predictions (orange x-marks), the interpolating parabola through those three points (blue curve), and the observations (red line). Each row shows a regional metric, and each column shows a tunable parameter. (The tunable parameters are defined in Table B1.) Many regions show a strongly nonlinear dependence on parameter value.

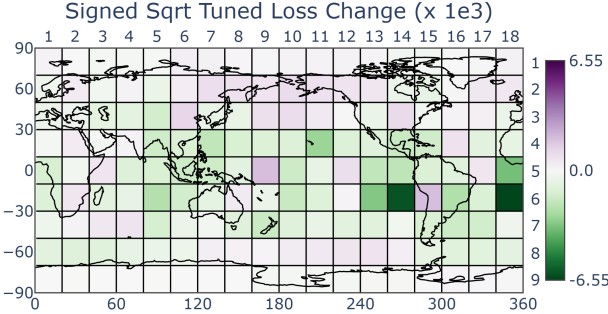

**Figure 5.** The change in loss function upon tuning, as given by Eq. (A4). Green indicates a reduced bias, and purple indicates a worsened bias. QuadTune thinks that it can diminish the stratocumulus biases (6_14 and 6_18), but it believes that the reduction comes only at the cost of worsening biases elsewhere, e.g., over Eastern United States (box 3_14) and over the Southern Hemisphere storm track (box 8_13).

Because of the nonlinearity in our example tuning run, we cannot predict QuadTune's behavior based solely on an analysis of the (linear) sensitivity matrix, $S_{ij}$. Instead, we will proceed to analyze the quasi-linear sensitivity matrix, $S_{ij}^+(\delta p_{opt})$, which was defined in Eq. (29). However, this matrix is available only after QuadTune has been run in order to obtain $\delta p_{opt}$. Our analysis merely aims to explain, after the fact, why QuadTune did what it did.

### 7.3 Which regional biases does QuadTune prioritize for removal?

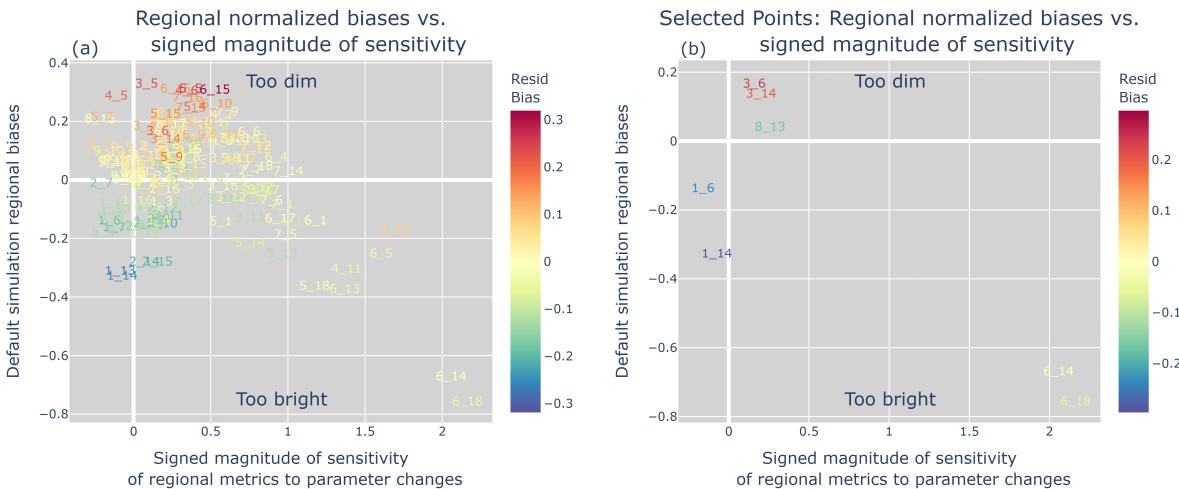

**Figure 6.** Regional biases $\delta b$ versus signed sensitivity, which is defined in Eq. A6. The color coding indicates the residual bias, which is defined in Eq. 12. Panel a) shows all regions in order to provide a broad-brush overview; panel b) shows only the specially designated regions, for clarity. The large stratocumulus biases (6_14 and 6_18) can be diminished by tuning because those regions are sensitive to parameter perturbations. The biases in insensitive regions (e.g., the Arctic region 1_14) cannot be budged, i.e., are stubborn.

Recall that we have defined a "tuning trade-off" as the situation in which tuning increases (i.e., worsens) the loss function in one region in order to improve it in another region (see Eq. 14). A map of QuadTune's predicted loss function (Figure 5) shows that QuadTune thinks that it can reduce the loss function in some regions (green), while leaving it unchanged in other regions (white) and worsened in other regions (purple). E.g., QuadTune strives to reduce the bias in the stratocumulus regions (6_14 and 6_18) at the expense of other regions (e.g., 3_14, 1_6, and 3_6).

Why does QuadTune prioritize bias reduction in the stratocumulus regions? In our tuning example, QuadTune appears to prioritize bias reduction where the loss is large and the magnitude of the sensitivity is large. Consider Figure 6, which plots the bias versus sensitivity for each region. Among all regions, the stratocumulus regions (6_14 and 6_18) have both the largest-magnitude bias *and* the largest sensitivities. The large sensitivities mean that parameter adjustments have the possibility of reducing those losses. The large losses mean that if the losses are indeed removed in those regions, then the reduction to the overall loss function will be relatively large. To reduce these large losses, the tuner sacrifices other regions that would lead to smaller gains. This leads to stubborn biases and tuning trade-offs.

The loss in stratocumulus regions is especially large because the loss function is based on mean squared error (27), rather than mean absolute error (MAE). However, when MAE is used instead, the optimal parameter values remain qualitatively similar (not shown), presumably because even with MAE, the stratocumulus regions dominate the error.

## 7.4 Which parameters does QuadTune adjust in order to remove the biases?

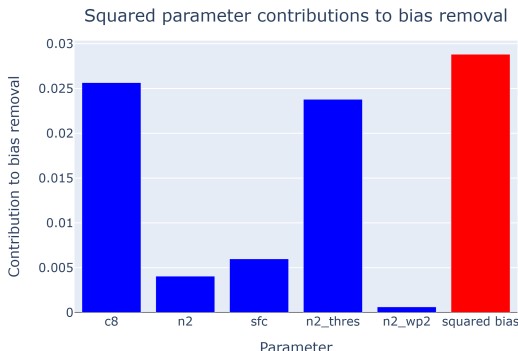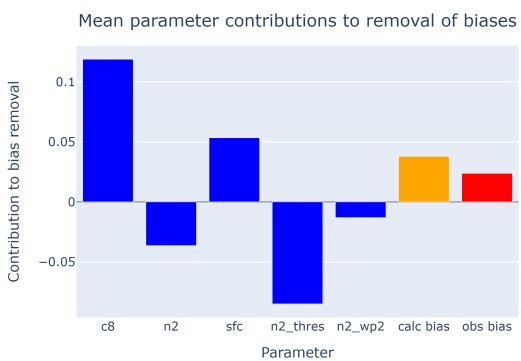

**Figure 7.** Contribution due to each parameter and the bias averaged over all metrics. The upper panel shows the square of metrics perturbations (A12). The lower panel shows the straight average (A13), plus QuadTune's estimate of bias of the tuned run, plus the observed bias from the default simulation. Although c8's sensitivity has little spatial correlation with the bias pattern, c8 is an important parameter because it restores global radiative balance.

It is helpful to know the relative influence of the tunable parameters because that suggests the relative influence of the terms in the model equations that contain those parameters. In addition, the parameter influence indicates which parameters could be

dropped from a subsequent tuning run. However, the influence is not simply related to the correlation between the parameter
and the bias.

QuadTune's recommended parameter values were plotted as the orange x-marks in Fig. 4. However, the size of the parameter perturbation doesn't necessarily indicate the effectiveness of that parameter in removing biases. A large parameter perturbation might have little effect on the biases, if the model is insensitive to that parameter.

One simple measure of the $j$th parameter's influence is the sum of a squared parameter perturbation over all $N$ regional metrics (where $T_{ij}^+$ is defined in Eq. 31):

$$\sum_{i=1}^{N} \left(T_{ij}^+\right)^2. \tag{32}$$

This sum comprises some terms in the (unweighted) loss function (27). This sum is plotted in Fig. 7. We see that, by this measure, for our example tuning run, the most influential parameters are n2_thresh and c8, followed by sfc and n2 (see the definitions of these parameters in Table B1).

Another way to measure the influence of parameters is singular value decomposition (SVD). (SVD has been widely used to decompose various types of matrices associated with PPEs, see, e.g., Dagon et al. (2020)). The first singular vector of the quasi-linear sensitivity matrix $\mathbf{S}^+$ contains a large fraction of the explained variance ($R^2 = 0.88$). The spatial pattern of the first left singular vector ($U_1$, Eq. A9, Figure 8) correlates fairly well with the spatial pattern of major biases (Fig. 3(a)).

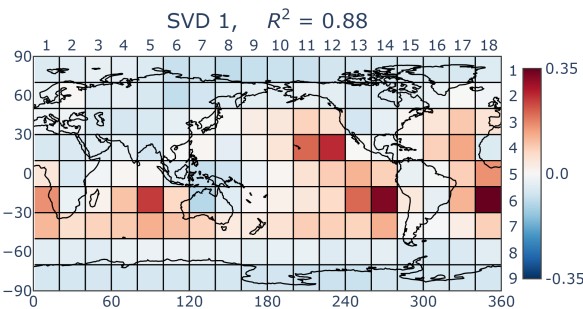

**Figure 8.** The first left singular vector, $U_1$ of the quasi-linear sensitivity matrix $\mathbf{S}^+$ (A9), arranged on a map. Most of the bias reduction is achieved by SVD 1. This is in part because of the strong projection onto the stratocumulus regions 6_14 and 6_18.

The first right-singular vector shows that the largest component is n2_thresh (Figure 9). The other major contributors are n2 and c8. The importance of c8 might seem counterintuitive (Elsaesser et al., 2024), given the low correlation of its sensitivity pattern (Figure 10) with the default bias pattern (Figure 3(a)). Quantitatively, the correlation between the sensitivity of c8 and the default bias is low (0.07), as measured by the parameter-bias correlation matrix (Figure 11).

Why, then, is c8 so important? It is because c8 is needed to restore global radiative balance (Elsaesser et al., 2024). QuadTune uses n2_thresh and n2 to reduce the local stratocumulus biases (Fig. 10(a),(b)), but the net effect of tuning those parameters is to dim the globe too much. Increasing c8 brightens the clouds in a more globally uniform way, without entirely undoing the

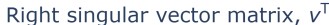

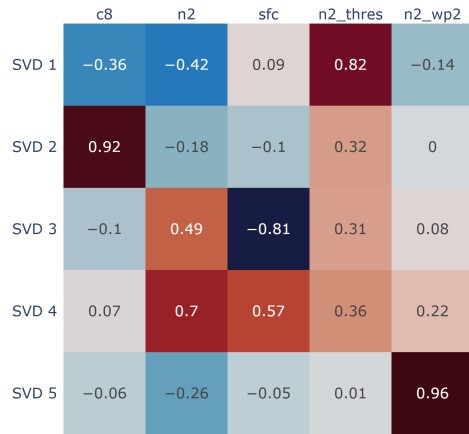

**Figure 9.** The right singular vector matrix, $\mathbf{V}^T$ of the quasi-linear sensitivity matrix $\mathbf{S}^+$ (A9). The first left singular vector, $\mathbf{V}_1^T$, which is the top row of $\mathbf{V}^T$, is dominated by n2_thresh.

benefits in the stratocumulus regions of tuning n2_thresh and n2. For a similar reason, the sfc parameter is also more important (Fig. 7) than what one might expect given its low correlation with the bias pattern (Fig. 11).

### 7.5 Biases that QuadTune fails to remove: Tuning trade-offs, stubborn biases, and the complexity introduced by nonlinear parameter dependence

The stratocumulus regions 6_14 and 6_18 have the largest biases among all regions. These biases contribute the most to the loss function, and hence, QuadTune is incentivized to remove them, if possible. And, in fact, QuadTune believes that it can remove those biases (Fig. 3(b)). QuadTune believes this because the parameter sensitivities of 6_14 and 6_18 are large (Fig. 6). Therefore, speaking in a broad-brush overview, QuadTune adjusts the parameter values primarily so as to remove those biases.

Other regions with smaller biases are de-prioritized by QuadTune. If those regions have biases and sensitivities that are
500 "consistent" with the stratocumulus regions, then those regions' biases will be reduced. But sometimes those regions are inconsistent, and then there is a trade-off between tuning away the stratocumulus biases and tuning away other biases. Because the other biases and sensitivities are weaker, they are sacrificed.

Our tuning example exhibits two notable types of tuning trade-offs between the stratocumulus regions and other regions:

1. *Regions with positively correlated sensitivity, but the "wrong" bias.* Consider the Eastern United States, specifically
region 3_14. Region 3_14 has a similar sensitivity to all the parameters as does 6_14 (or 6_18) (see the matrix-equation bar chart in Fig. 12). However, 3_14's bias has the opposite sign. Whereas 6_14 is too bright, 3_14 is too dim. In whatever

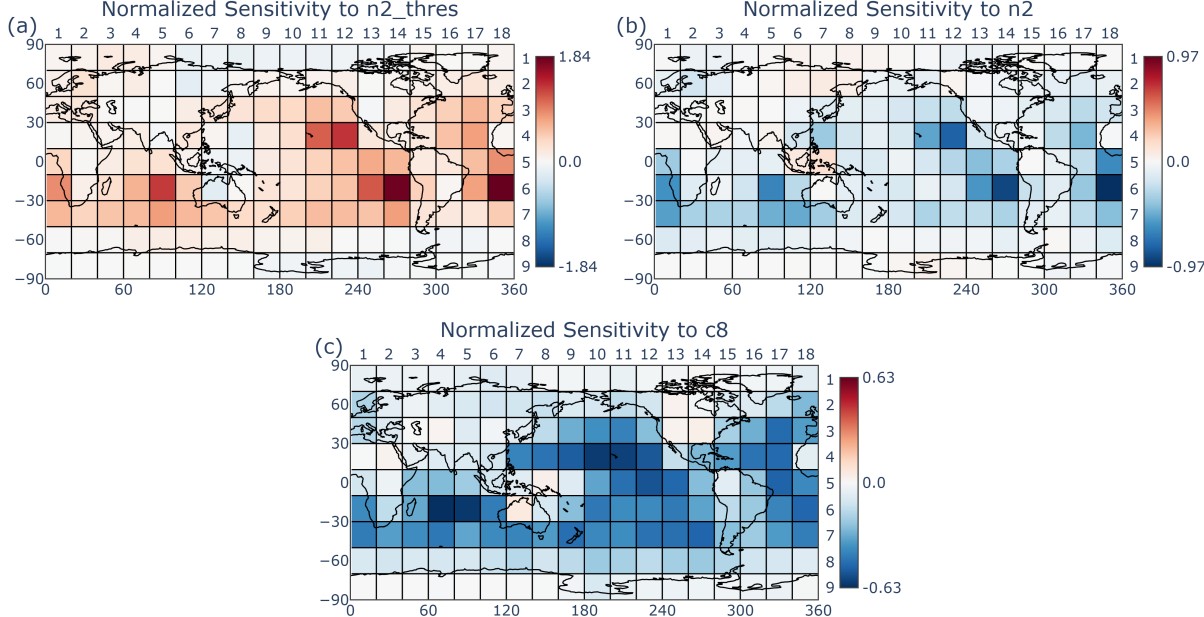

**Figure 10.** Map of normalized sensitivity, $\mathbf{S}_j^+$, of a) n2_thres, b) n2, c) c8. The spatial pattern of SVD 1 (Fig. 8) resembles the sensitivity of two of its strongest components, n2_thresh and n2, but not c8, despite c8's importance (Fig. 9).

way QuadTune adjusts the parameter values, improving 6_14 will necessarily worsen 3_14. In our example, points such as 3_14 appear in the upper-right quadrant of the bias-sensitivity scatterplot, opposite the lower-right quadrant where the large biases such as 6_14 appear (Fig. 6). (For a toy example of this trade-off, see Eq. (16) and the related discussion.)

2. *Regions with a "correct-sign" bias, but with an anti-correlated sensitivity.* Consider region 1_6, which is north of Siberia. Its bias has the same sign as the stratocumulus biases (it's too bright), but its response to, for instance, parameter n2_thresh has the opposite sign to that of the stratocumulus regions (Fig. 12). Responses with different signs in different regions have been noticed, e.g., by Qian et al. (2024). The reason for the different response in 1_6 is that the dependence on n2_thresh is strongly nonlinear (Fig. 4). Increasing n2_thresh dims the Sc regions, as desired, but

brightens 1_6, unfortunately. Such points appear in the lower-left quadrant of the bias-sensitivity scatterplot (Fig. 6).

    3. *Overcorrection.* Sometimes a region starts with a bias of one sign, but after tuning, it is left with a bias of the opposite sign. An example is Region 8_13, in the SH storm track. The clouds in 8_13 are originally too dim, but they end up too bright (Fig. 12). The over-brightening occurs because the dependence on n2_thresh is nonlinear and because the sensitivity to c8 is unusually strong in this region (Fig. 4).

We now list two other types of bias that cannot be removed, regardless of their consistency or inconsistency with the prioritized Sc biases, 6_14 and 6_18.

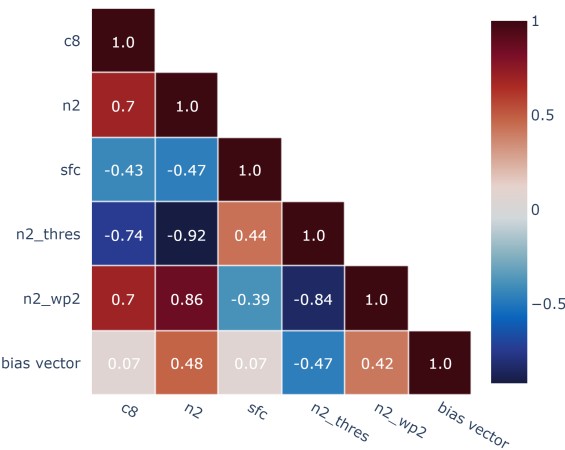

**Figure 11.** Quasi-linear correlations among parameters and the bias (A11). The parameters n2_thresh and n2 have the strongest correlation to the bias pattern, and c8 has a weak correlation.

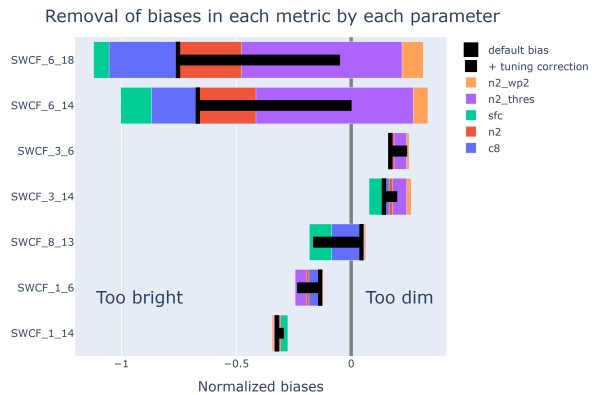

**Figure 12.** Visualization of the quasi-linear tuning matrix equation (A14). Each row of bars represents a row of the matrix equation (A14). The vertical black bars indicate the default bias value. The lengths of the horizontal black bars indicate QuadTune's prediction of removable bias. Each colored rectangle represents the change in a metric $i$ due to the $j$th parameter perturbation ($T_{ij}^+$). The plot illustrates, e.g., stubborn biases (1_14) and tuning trade-offs (e.g., 3_14).

1. *Stubborn bias.* A stubborn bias occurs in a region that has a non-negligible bias but has little sensitivity to any parameter (see Eq. 15). An example is Region 1_14, in the Arctic north of Canada. The large bias and small sensitivity of 1_14 is

**Table 1.** Root mean square error (RMSE) of SWCF of global EAM simulations.

| Simulation | RMSE (W m$^{-2}$) |
|---|---|
| Default (untuned) EAM parameters | 12.4 |
| Hand-tuned | 10.1 |
| Quad-tuned, $20° \times 20°$, 1-yr runs (default) | 10.4 |
| Quad-tuned, $30° \times 30°$, 1-yr runs | 10.25 |
| Quad-tuned, $20° \times 20°$, 2-yr runs | 10.55 |
| Piecewise-linear-tuned, $20° \times 20°$, 1-yr runs | 10.3 |
| Quad-tuned with interaction term, $20° \times 20°$, 1-yr runs | 10.1 |

The top three rows use 1-year sensitivity simulations and $20° \times 20°$ regions, but have different methods of tuning (see Fig. 2). The fourth and fifth rows list two alternative quad-tuned configurations, one of which uses $30° \times 30°$ regions and the other of which uses sensitivity simulations that last two years. The sixth row replaces the quadratic emulator with a piecewise-linear emulator (33). The seventh row includes a single interaction term in the quadratic emulator (34). Modifying the size of regions, duration of sensitivity simulations, or shape of the emulator has only modest impacts.

evident in both the matrix-equation bar chart (Fig. 12) and the three-dot plot (Fig. 4). The large bias and small sensitivity also means that 1_14 resides on or near the y-axis of the bias-sensitivity scatterplot, and far from the x-axis, which has zero bias (Fig. 6(b)).

Possibly 1_14's bias is stubborn to perturbations to CLUBB's parameters because the clouds in 1_14 are impacted less by CLUBB's tendencies than by tendencies of other parameterizations, such as microphysics. Because of 1_14's lack of sensitivity, removing its bias via CLUBB would require a large adjustment to the parameter values. Such a large parameter adjustment would over-perturb other more sensitive regions, worsening the overall fit. Therefore, QuadTune leaves the stubborn bias in 1_14 unimproved.

2. *Nonlinear zugzwang.* In this situation, the default parameter value is the best possible value because the quadratic parameter dependence prevents any parameter perturbation from improving the bias. An example is Region 3_6, which is located in China. The dependence of SWCF in 3_6 on each parameter is parabolic (Fig. 4). Because each parabola curves away from the observed value of SWCF, any large parameter perturbation worsens the fit. This worsening is "internal" to region 3_6, rather than a tuning trade-off with other regions.

If a QuadTune user wishes to remove a residual bias in a particular regional metric that remains after tuning, then he must either find another global-model parameter to tune, or else make a model structural change, or else upweight the region.

## 8 How sensitive are results to the tuning configuration and emulator functional form?

The sensitivity of the quad-tuned results to various configurations of QuadTune is listed in Table 1. All simulations in Table 1 have a duration of 5 years, even if the parameter-value recommendations are based on 1- or 2-year perturbed simulations. The default quad-tuned configuration produces a RMSE of SWCF ($10.4$ W m$^{-2}$) that is better than the untuned EAM run ($12.4$ W m$^{-2}$) but somewhat worse than the hand-tuned run ($10.1$ W m$^{-2}$).

The quality of the tuning results is moderately sensitive to the size of the tuning regions and the duration of the sensitivity
simulations. In the default tuning configuration, the metrics are averaged over $20° \times 20°$ regions, and the perturbation global simulations last 1 year. If the regions are coarsened to $30° \times 30°$, then the RMSE actually improves slightly ($10.25$ W m$^{-2}$). If the regions are $20° \times 20°$, but the perturbation simulations are lengthened to 2 years, then the RMSE worsens slightly ($10.55$ W m$^{-2}$). The sense of these changes is counter-intuitive, but the changes are modest, and we believe that they are within the range of random error.

The sensitivity of the RMSE to the emulator shape is addressed with two independent experiments. In the first, we replace the quadratic emulator (22) with a piecewise linear emulator:

$$
m_i(\boldsymbol{p}) = m_i(\boldsymbol{p}_{def}) + \sum_{j=1}^{P} \begin{cases} \left.\dfrac{\partial m_i}{\partial p_j}\right|_{\text{left}} \delta p_j & \text{if } p_j < p_{j,def} \\ \left.\dfrac{\partial m_i}{\partial p_j}\right|_{\text{right}} \delta p_j & \text{if } p_j \geq p_{j,def} \ . \end{cases}
\tag{33}
$$

In this equation, we have assumed that the default parameter value lies between the high and low values. "Left" denotes the slope between the default and low value, and "right" denotes the slope between the default and high value. The RMSE of SWCF
changes from $10.4$ W m$^{-2}$ to $10.3$ W m$^{-2}$, demonstrating little sensitivity to the emulator shape in this particular example. In the second experiment, we add a single interaction term to (22), namely

$$
\frac{1}{2} \frac{\partial^2 m_i}{\partial p_1 \partial p_2},
\tag{34}
$$

where $p_1 = $ c8 and $p_2 = $ n2_thresh. This interaction term was chosen because QuadTune perturbs both these parameters a lot in the stratocumulus regions, 6_14 and 6_18. Hence we expect the interaction term between these two parameters to be
large as well. Ideally, of course, one might like to include all interaction terms, but that would require 10 extra simulations. However, even including one interaction term improves the RMSE of SWCF from $10.4$ W m$^{-2}$ to $10.1$ W m$^{-2}$, which equals the hand-tuned RMSE. Once again, the result gives a sense of the sensitivity to emulator shape.

We know that parametric error remains in QuadTune's default-configuration solution because that solution has greater RMSE ($10.4$ W m$^{-2}$) than both the hand-tuned and interaction solutions ($10.1$ W m$^{-2}$). Given the existence of parametric error even
after tuning, the structural errors have not been fully isolated. Because of this, the model error that we analyze is in fact a mixture of structural error and parametric error.

This is a vexing problem in general because, no matter how sophisticated one makes an emulator, one can never know if parametric error has been eliminated. There may always exist an unexplored pocket of parameter space that produces a much

lower error. However, in our example, varying the emulator and hand tuning both produce a RMSE of SWCF between 10 and 11 W m$^{-2}$. In contrast, eliminating all parametric and structural (and observational) error would, in theory, produce a RMSE of 0 W m$^{-2}$. If we assume that fully eliminating parametric error does not reduce RMSE much below 9 or 10 W m$^{-2}$, then the model error is still dominated by structural error. If so, our analysis of structural error remains qualitatively useful, despite the admixture of post-tuning parametric error.

## 9 Caveats and future work

At present, QuadTune is a barebones tuner. It could be extended in many ways. We list several of them now.

1. *Calculate an ensemble of perturbed parameter sets.* By calculating multiple near-optimal parameter sets, a calibrated physics ensemble can be created, as in Elsaesser et al. (2024). Such an ensemble of parameter sets would allow us to construct error bars on parameter values, regional metrics, and other quantities. Calculating an ensemble is particularly important because the quartic loss function potentially has multiple minima (Section 5.2).

2. *Find a better way to choose which parameters to tune (i.e., do feature selection).* Currently, we are manually pruning away unimportant parameters simply by running QuadTune and deleting parameters that contribute little, as measured by, e.g., Fig. 7(a). However, it would be preferable to have a more efficient and objective method.

3. *Simultaneously tune multiple fields.* In this initial exposition, we tuned only one metric (SWCF). However, tuning only one metric is prone to overfitting the tuned metric at the expense of other metrics. Therefore, we should explore the question of how much data is needed to avoid overfitting. Tuning multiple metrics also requires design choices, such as how to weight different metrics appropriately.

4. *Explore the effects of parameter interactions.* The accuracy of our emulator (22) is improved by including interactions between parameters, especially those that are strongly perturbed. However, doing so requires performing extra global simulations. The benefits and costs should be explored further.

5. *After quad-tuning once, systematically iterate in order to improve optimal parameter estimates.* QuadTune could be used to carry out an initial, exploratory stage of tuning in order to prepare for later stages of more refined tuning. If desired, each later stage could also be carried out by QuadTune, but with an updated default simulation and an updated set of sensitivity simulations. This would allow the user to sequentially run small batches of $2P+1$ simulations until adequate parameter values are found. For example, as an initial exploratory experiment, we have started with the iteration-1 optimal parameters from QuadTune (which produce RMSE = 10.4 W m$^{-2}$), constructed a new batch of $2P+1$ simulations with reduced parameter perturbations about the iteration-1 optimum, re-quad-tuned once, and found an improved RMSE of 10.0 W m$^{-2}$ (not shown). Alternatively, the later stages could employ a more sophisticated tuning method than Quad-Tune. In this case, the first stage (i.e., quad-tuning) would help demarcate reasonable ranges of the parameter values, i.e. would help do feature selection for later stages of iteration (Hastie et al., 2009).

# 10 Conclusions

QuadTune provides software to tune away parametric error — approximately but quickly — in global atmospheric models. It reduces the required number of global sensitivity runs, which are expensive, by assuming a particularly simple emulator of parameter dependence, namely, a quadratic, non-interacting one (Eq. 22).

QuadTune does not attempt to find the true global optimal parameter set, but merely finds the approximate location of a nearby minimum in parameter space. Nevertheless, the tuner is capable of removing much of the parametric bias that is removable in hand tuning. E.g., in our example, the RMSE in SWCF is reduced from 12.4 W m$^{-2}$ to 10.4 W m$^{-2}$, as compared to a hand-tuned value of 10.1 W m$^{-2}$. In this particular example, this reduction requires only 11 global tuning runs (although the important parameters and their ranges must be identified beforehand). Based on our experience thus far, we are hopeful that QuadTune will prove useful for quick retuning after a structural model change.

In our example tuning analysis, we find that nonlinearity in the parameter dependence cannot be ignored (Fig. 4). Therefore, for many of our parameters, it is unhelpful to think simply in terms of a single sensitivity. To help understand some of the non-linear effects, we base our diagnostics on a quasi-linear sensitivity matrix, $S_{ij}^{+}$ (Eq. 29). However, $S_{ij}^{+}$ is a function of QuadTune's estimate of optimal parameter values, and hence dependence on $S_{ij}^{+}$, restricts us to after-the-fact interpretation of QuadTune's behavior.

One of our goals is explainability. Namely, we wish to understand why QuadTune finds the parameter values that it does and what prevents QuadTune from reducing biases further. QuadTune's explainability is aided by its simple, quadratic emulator and also by its diagnostic plots that, for instance, compress a multi-variate regression problem to a univariate scatterplot (Fig. 6) and visualize the tuning equation (30) with a decorated bar chart (Fig. 12). The compression in the scatterplot is achieved by calculating the overall sensitivity to all parameters rather than retaining the individual sensitivity to each parameter. The bar chart visualizes the contributions of each parameter perturbation to removing each regional bias, but more generally, it provides a way to visualize any (small) matrix equation. The mathematical quantities plotted in QuadTune's diagnostics are listed in Appendix A.

In our example tuning run, how does QuadTune adjust parameter values in order to reduce the biases? QuadTune perturbs two parameters that are strongly correlated with the bias (n2_thresh and n2) and also perturbs another parameter that is weakly correlated (c8) (see Fig. 9, and Section 7.4). Even though c8 is weakly correlated, an adjustment to c8 is needed to restore radiative balance after the adjustments to n2_thresh and n2 (Fig. 7).

We encountered two classes of bias that could not be removed by tuning (Section 7.5). The first class is the class of stubborn biases, as defined in Eq. (13) and illustrated in Eq. (15). A stubborn bias is a relatively large-magnitude bias with a relatively small sensitivity to parameter perturbations. We found, for instance, that there is a stubborn bias in the Canadian Arctic region 1_14 (Fig. 6). The second class of biases that we encountered is the class of tuning trade-offs. We define a tuning trade-off as a sacrifice of the loss function in one region in order to improve the loss function in another region (see Eq. 14, Eq. 16, and Fig. 5). A tuning trade-off may involve, for example, two regions with the same sensitivity but different biases (see Region

3_14 in the United States in Fig. 12). Alternatively, it may involve the same sign of bias but a different sensitivity (e.g., the Siberian Arctic Region 1_6 in Fig. 12).

QuadTune can tell us which parameters matter in which regions. This information could, in principle, provide hints about model structural error if a parameter could be associated with a particular term (i.e., process) in a budget. However, QuadTune itself does not analyze how parameters enter global-model equations. That requires further analysis.

*Code availability.* The EAM global-model source code, including CLUBB, is located at https://github.com/larson-group/v3atm/releases/tag/quadtune_2025_paper and is archived on Zenodo (see Edwards et al., 2025). The QuadTune Python scripts are available at https://github.
com/larson-group/quadtune and are archived on Zenodo (see Larson et al., 2025).

## Appendix A: Mathematical basis of diagnostics plots

This appendix lists the mathematical quantities that are plotted in QuadTune's diagnostic plots.

### A1   Three-dot plot (Figure 4)

The purpose of this plot is to give detailed information about the sensitivity of selected regions to each parameter. In particular,
the plot conveys the degree of nonlinearity in the parameter dependence.

    For a given regional metric and parameter, the three black "dots" plot the simulated value of the metric (here, SWCF) versus the parameter value for the default simulation and the two sensitivity runs that perturb the parameter high and low. The blue line is the parabola that uniquely interpolates these three dots. For the regional metric $m_i$ and parameter $p_j$, it is

$$m_i(p_j; p_{j,def}) = m_i(p_{j,def}) + \frac{\partial m_i}{\partial p_j} \delta p_j + \frac{1}{2} \frac{\partial^2 m_i}{\partial p_j^2} (\delta p_j)^2, \tag{A1}$$

where $\delta p_j \equiv p_j - p_{j,def}$. This is essentially one term in the quadratic emulator (Eq. (22)).

### A2   Loss map (Figure 5)

The contribution to the loss function in the $i$th region is

$$L_i(\delta \boldsymbol{p}) \equiv \sigma_i^2 \left[ -\delta b_i - \sum_{j=1}^{P} \left( \frac{\partial m_i}{\partial p_j} \delta p_j + \frac{1}{2} \frac{\partial^2 m_i}{\partial p_j^2} (\delta p_j)^2 \right) \right]^2. \tag{A2}$$

Of particular interest is the change in loss upon tuning (tuned minus default):

$\delta L_i \equiv L_i(\delta \boldsymbol{p}_{opt}) - L_i(0). \tag{A3}$

When $\delta L_i < 0$, the $i$th bias is reduced in magnitude, and when $\delta L_i > 0$, the bias is worsened.

To produce better color contrast, Fig. 5 plots the quantity

$$1000 \, \mathrm{sgn}(\delta L_i) \sqrt{\delta L_i}. \tag{A4}$$

## A3   Bias-sensitivity scatterplot (Figure 6)

Here our goal is to compress information about a multiple linear regression problem into a familiar univariate scatterplot. I.e., we wish to create a mock univariate scatterplot.

To do so, we create a scatterplot of the bias $\delta b_i$ versus the signed sensitivity of each regional metric, $\mathrm{signedSens}_i$. The sign of $\mathrm{signedSens}_i$ is positive for any region whose sensitivity row vector is positively correlated with the sensitivity row vector of the most sensitive region. Likewise, the sign is negative for negatively correlated regions.

The overall sensitivity of the $i$th region to all parameters, $\mathrm{sens}_i$, is taken to be the Euclidean norm of the sensitivity due to all parameters:

$$\mathrm{sens}_i = \sqrt{\sum_j (S_{ij}^+)^2} \tag{A5}$$

where $S_{ij}^+$ is defined in Eq. (29). We plot a signed version of $\mathrm{sens}_i$, $\mathrm{signedSens}_i$:

$$\mathrm{signedSens}_i = \mathrm{signOfSens}_i \mathrm{sens}_i \quad \text{(no sum over i)}. \tag{A6}$$

Here, $\mathrm{signOfSens}_i = 1$ if the sensitivity row-vector of the $i$th region is positively correlated with the sensitivity row-vector of the most sensitive region. On the other hand, $\mathrm{signOfSens}_i = -1$ if it is negatively correlated.

We assume that QuadTune prioritizes the reduction of the bias of the region with the greatest sensitivity. We denote that region $i = I$:

$$I = \arg\max_i (\mathrm{sens}_i) \tag{A7}$$

Moreover, $\mathrm{signOfSens}_i$ is simply the sign of the dot product between row $i$ and row $I$ of $S_{ij}^+$:

$$\mathrm{signOfSens}_i = \mathrm{sign} \left( \sum_j S_{ij}^+ S_{Ij}^+ \right) \tag{A8}$$

## A4 Singular vectors (Figures 8 and 9)

$$\left[ \quad \boldsymbol{S}^{+} \quad \right] \approx \sigma_1 \left[ \boldsymbol{U}_1 \right] \left[ \quad \boldsymbol{V}_1^{T} \quad \right] + \ldots \tag{A9}$$

Recall that, for $S_{ij}^{+}$, the $i$th row corresponds to the $i$th regional metric, and the $j$th column corresponds to the $j$th parameter. The SVD separates the dependence on regions and parameters, so that the column vector $\boldsymbol{U}_1$ represents the spatial pattern and the row vector $\boldsymbol{V}_1^{T}$ represents the parameter dependence. Figure 8 color-codes the values of the elements of $\boldsymbol{U}_1$ and arranges them on a gridded map. Figure 9 color-codes the entire $\boldsymbol{V}^{T}$ matrix. The first row corresponds to the first singular vector.

## A5 Parameter-correlation matrix (Figure 11)

First we extend the quasi-linear sensitivity matrix $\boldsymbol{S}^{+}$ by appending the bias column vector:

$$\boldsymbol{SE}^{+} \equiv \left[ \boldsymbol{S}^{+}{}_{(N \times P)} \quad \delta\boldsymbol{b}_{(N \times 1)} \right] \tag{A10}$$

Then we debias and normalize the columns of $\boldsymbol{SE}^{+}$ in order to form the matrix $\boldsymbol{SE}_n^{+}$. Finally, the correlation matrix is given by:

$$\boldsymbol{C} \equiv \boldsymbol{SE}_n^{+T} \boldsymbol{SE}_n^{+} \tag{A11}$$

One may interpret $\boldsymbol{C}_{ij}$ as $\cos(\theta_{ij})$, where $\theta_{ij}$ is the angle between two columns of $\boldsymbol{SE}^{+}$.

## A6 Parameter contribution bar chart (Figure 7)

In Figure 7(a), which shows the absolute value of each contribution by an individual parameter to all regions, the height of the $j$th bar (parameter) is given by:

$$\text{bar height}_j = \sum_{i=1}^{N} \left| T_{ij}^{+} \right| \tag{A12}$$

In Figure 7(b), which shows the mean contribution of each parameter,

$$\text{bar height}_j = \sum_{i=1}^{N} T_{ij}^{+} \tag{A13}$$

## A7 Matrix-equation bar chart (Figure 12)

This bar chart visualizes the matrix equation

$$\delta b_i = \sum_{j=1}^{P} T_{ij}^{+} + \delta b_{resid,i} \tag{A14}$$

Because the chart can be applied to any such matrix equation, it is quite general. Each element $T_{ij}^{+}$ is represented by a colored bar. Because every individual matrix element is displayed, the chart is a comprehensive depiction of the matrix equation. (However, in Fig. 12, only selected rows are shown.) The value of the bias $\delta b_i$ is represented by the location of the vertical black line. The residual bias $\delta b_{resid,i}$ is given by the distance between the end of the horizontal black line and the y-axis. The length of the horizontal black line represents the removable portion of the bias $\delta b_{remov,i} = \delta b_i - \delta b_{resid,i}$.

## Appendix B: How CLUBB parameters appear in CLUBB-taus equations

This appendix lists the terms that contain the five parameters that we tune in this paper (c8, n2_thresh, sfc, n2, and n2_wp2). The purpose of this equation sketch is mostly to give a flavor of some of the potential interactions between parameters. For ease of exposition, we ignore extra damping that CLUBB applies in clear, stable layers. We also ignore the fact that some of CLUBB's damping is reduced near the ground. For a fuller account of CLUBB's tau damping, see Guo et al. (2021) or Zhang et al. (2023).

The five parameters appear in damping time scales in equations for three of CLUBB's prognosed subgrid turbulence moments:

$$\frac{\partial \overline{w'^2}}{\partial t} = ... - \frac{\overline{w'^2}}{\tau_{w'^2}} \tag{B1}$$

$$\frac{\partial \overline{w'^3}}{\partial t} = ... - C_{11} \frac{3g}{\theta_0} \overline{w'^2\theta_v'} - \frac{c8}{\tau_{w'^2}} \overline{w'^3} \tag{B2}$$

$$\frac{\partial \overline{w'x'}}{\partial t} = ... - \frac{\overline{w'x'}}{\tau_{w'x'}}, \tag{B3}$$

where $\overline{w'^2}$ is the variance of vertical velocity, $\overline{w'^3}$ is the third-order moment of vertical velocity, and $\overline{w'x'}$ is the turbulent flux of either total moisture ($x' = r_t'$) or liquid water potential temperature ($x' = \theta_l'$). Here the $C_{11}$ term is a buoyancy damping term that competes with the more generic damping term, $-c8\,\overline{w'^3}/\tau_{w'^2}$, as discussed in Section 7.2. The damping on $\overline{w'^3}$ is assumed to be proportional to the damping on $\overline{w'^2}$, with a proportionality constant c8. In CLUBB, the ratio of $\overline{w'^3}$ to $\overline{w'^2}$ affects boundary layer depth and cloud brightness (Ma et al., 2022).

The damping time scale of $\overline{w'^2}$, $1/\tau_{w'2}$, is the sum of two other damping time scales that attempt to model the effects of physical conditions such as the degree of stable stratification, as measured by the Brunt-Väisälä frequency, $N$:

$$\frac{1}{\tau_{w'2}} = \frac{1}{\tau_{\text{noN}}} + \frac{1}{\tau_{\text{N,wp2}}}. \tag{B4}$$

The damping on $\overline{w'x'}$ adds an extra factor in order to better account for the strong effects of stable stratification on cloud-top entrainment fluxes (Guo et al., 2021):

$$\frac{1}{\tau_{w'x'}} = \left(\frac{1}{\tau_{\text{noN}}} + \frac{1}{\tau_{\text{N}}}\right)$$
$$\times \left(1 + C_{i\tau\text{wpxpRi}} Ri_g^{0.5} H(N^2 - N_{\text{thresh}}^2)\right). \tag{B5}$$

Here, $Ri_g$ is the gradient Richardson number, $C_{i\tau\text{wpxpRi}}$ is a constant, and $H$ is the Heaviside step function, which equals 1 if its argument is positive and 0 if its argument is negative. Also, the expression contains a tunable parameter $N_{\text{thresh}}^2$ (also denoted n2_thresh). The parameter $N_{\text{thresh}}^2$ is a threshold below which the extra damping factor is shut off.

The damping is neutrally stratified layers ("noN") is modeled by

$$\frac{1}{\tau_{\text{noN}}} = \underbrace{C_{i\tau\text{sfc}} \frac{u_*}{\kappa z}}_{\text{surface}}$$
$$+ \underbrace{C_{i\tau\text{shear}} \sqrt{\left(\frac{\partial \overline{u}}{\partial z}\right)^2 + \left(\frac{\partial \overline{v}}{\partial z}\right)^2}}_{\text{shear}}$$
$$+ \underbrace{C_{i\tau\text{bkgnd}} \frac{1}{\tau_{\text{const}}}}_{\text{background}}. \tag{B6}$$

where $C_{i\tau\text{sfc}}$ (also denoted sfc) is a tunable parameter that governs the strength of damping near the ground, where $z$ is the altitude above ground, $u_*$ is the surface friction velocity, and $\kappa$ is the von Karman constant. The second term adds extra damping in sheared layers, and the last term provides background damping in unsheared layers aloft. For more details, see Guo et al. (2021).

To this neutrally stratified damping, we add extra damping to $\overline{w'x'}$ in stable layers:

$$\frac{1}{\tau_{\text{N}}} = C_{i\tau\text{N}} \max(0, N), \tag{B7}$$

where $C_{i\tau\text{N}}$ (also denoted n2) is another tunable parameter. To $\overline{w'^2}$, we add

$$\frac{1}{\tau_{\text{N,wp2}}} = C_{i\tau\text{N,wp2}} \max(0, N), \tag{B8}$$

**Table B1.** Definitions of tunable parameters.

| Parameter Name | Meaning | Low Value | Default Value | High Value | Eqn |
|---|---|---|---|---|---|
| c8 | Damping on $\overline{w'^3}$ | 0.3 | 0.5 | 0.9 | (B2) |
| n2_thresh | Lower $N^2$ threshold for extra damping | 2.0e-4 | 3.3e-4 | 5.0e-4 | (B5) |
| sfc | Lower surface damping | 0.0 | 0.3 | 0.4 | (B6) |
| n2 | Damping of $\overline{w'x'}$ for large $N^2$ | 0.2 | 0.65 | 1.0 | (B7) |
| n2_wp2 | Damping of $\overline{w'^2}$ for large $N^2$ | 0.0 | 0.2 | 0.4 | (B8) |

The low and high parameter values are the ones used in our $2P$ one-at-a-time sensitivity runs.

where $C_{i\tau N,\text{wp2}}$ (also denoted n2_wp2) is a tunable parameter.

The tunable parameters are summarized in Table B1.

Inspecting these expressions, one can find several potential parameter interactions. For instance, sfc, n2, and n2_thresh all appear in $1/\tau_{w'x'}$ (B5). If one of the three parameters has smaller effects than the others, then the model sensitivity may roll off, as mentioned in Section 7.2. To cite another example, sfc and n2_wp2 both appear in $1/\tau_{w'^2}$.

*Author contributions.* ZG and VL wrote the software, and VL drafted the manuscript. BS, CZ, YQ, and SX discussed the science and helped revise the manuscript. GD consulted on statistical methods.

*Competing interests.* The authors declare that they have no conflicts of interest.

*Acknowledgements.* We thank two anonymous reviewers for their suggestions. We acknowledge a helpful conversation with Andrew Yarger, Benjamin Wagman, Lyndsay Shand, and Gavin Collins. We are grateful to Chris Terai for facilitating supercomputer access. This research was supported as part of the Energy Exascale Earth System Model (E3SM) Science Focus Area, funded by the U.S. Department of Energy (USDOE), Office of Science, Office of Biological and Environmental Research Earth System Model Development program area. The work at the University of Wisconsin — Milwaukee and Lawrence Livermore National Laboratory (LLNL) was performed under the auspices of the U.S. DOE by LLNL under Contract DE-AC52-07NA27344. BS is grateful for support by grant DE-SC0025252 from the USDOE's Regional and Global Model Analysis program. GZ was supported by the National Natural Science Foundation of China (Grant 42175164) and the Basic Scientific Research Project of Institute of Atmospheric Physics during the 14th Five-year Plan period.

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
