# Peer review of "QuadTune version 1: A regional tuner for global atmospheric models"

_EGUsphere, 2025_

## Author Comment (AC1)

**Responses to reviews of "QuadTune version 1: A regional tuner for global atmospheric models"**

**Vincent E. Larson, Zhun Guo, Benjamin A. Stephens, Colin Zarzycki, Gerhard Dikta, Yun Qian, Shaocheng Xie**

We thank both reviewers for taking the time to provide thoughtful comments.

The reviewer comments are repeated below in blue italics, with responses interspersed in a non-italicized font. Changes to the manuscript text appear in orange.

**1   Responses to Reviewer 1**

*The paper presents QuatTune, a software that can be use to do multi-parametric calibration of climate models. The paper is well written and clearly structured, the presetation is clear and methodology explaind in detail and with useful pedagogical descriptions. The method is applied to a development version of E3SM. I have thoroughly enjoyed reading the paper, and I would recommend publication afer minor revisions. Please see my specific comments below.*

Thank you.

*Figure 3. I think it would be worth reiterating in the caption that the midpoint of the parabola is the default parameter value. It would be interesting to see the simulated values for these regions, in addition to the QuadTune optimised predictions.*

The caption now notes that the middle dot is the default parameter value:

"Three-dot plot showing the SWCF values from the sensitivity and default runs (three black dots, with the middle dot being the default)"

We decided that including simulation values from the EAM global model would make the plot too busy, and so we left them out.

*L441-2. The los function is based on least squares, which prioritises regions with large biases. I´d like to see a brief explanation on the effect of choosing a different funtional form of the los function. Is this easily configurable in QuadTune?*

Thanks for the good idea. We tried MAE instead of MSE, and it didn't make a big difference. The revised manuscript now contains the sentences:

"The loss in stratocumulus regions is especially large because the loss function is based on mean squared error, rather than mean absolute error (MAE). However, when MAE is used instead, the optimal parameter values remain qualitatively similar (not shown), presumably because even with MAE, the stratocumulus regions dominate the error."

*I don´t think the computational cost of QuadTune (the optimisation process) is described in the paper. What is it? How does it scale with respect to parameters and target metrics?*

The introduction now includes the sentence:

"Once the global runs are completed, using QuadTune to find optimal parameter values takes only seconds on a laptop computer."

Also, Section 5.2 now states:

"For a quadratic function, Powell's method requires $P(P + 1)$ 1D minimizations. It scales linearly with the number of regional metrics."

**2 Responses to Reviewer 2**

*The paper approaches the tuning problem in a way that aims to use as few expensive runs as possible, in order to learn about structural error / parametric uncertainty / biases.*

Thank you for your review.

Yes, you have hit upon a central goal of the manuscript: to reduce the cost of tuning. The standard paradigm these days requires creating a PPE with > 100 global large-scale simulations, but this cost is unaffordable for some parameterization developers. We want the most accurate emulator that we can afford, but increasing the emulator accuracy inevitably costs more, and that cost cannot be ignored in some practical applications. This practical consideration of cost motivates our alternative paradigm.

*I'm not currently convinced that all assumptions have been justified (e.g., ignoring errors of emulator) and I'm not wholly convinced that the method is working well based on the current example, however I think these could be fixable with additional justifications, and perhaps an example where it works*

*better, and additional exploration of sensitivities in the method due to assumptions being made:*

We have added a justification of the use of QuadTune's emulator in Section 8:

"We know that parametric error remains in QuadTune's default-configuration solution because both the hand-tuned and interaction solutions have lower RMSE (10.1 W m$^{-2}$) than do QuadTune's default solution (10.4 W m$^{-2}$). Given the existence of parametric error even after tuning, the structural errors have not been fully isolated. Because of this, the model error that we analyze is in fact a mixture of structural error and parametric error.

This is a vexing problem in general because, no matter how sophisticated one makes an emulator, one can never know if parametric error has been eliminated. There may always exist an unexplored pocket of parameter space that produces a much lower error. However, in our example, varying the emulator and hand tuning both produce a RMSE of SWCF between 10 and 11.4 W m$^{-2}$. In contrast, eliminating all parametric and structural (and observational) error would, in theory, produce a RMSE of 0 W m$^{-2}$. If we assume that fully eliminating parametric error does not reduce RMSE much below 9 or 10 W m$^{-2}$, then the model error is still dominated by structural error. If so, our analysis of structural error remains qualitatively useful, despite the admixture of post-tuning parametric error."

In Section 8, we present two additional experiments on sensitivities to emulator shape. Namely, we have added an interaction term, and replaced the quadratic emulator with a piecewise-linear one. These give a sense of the size of the emulator error. These experiments required a laborious extension of the QuadTune code base, as can be seen in the public QuadTune GitHub repository.

The example tuning run that the manuscript presents is a good one because it illustrates the tuning quality that can be expected from QuadTune and the insights that can be gained. It is a real example that arose in the course of developing a global atmospheric model.

*1. The emulator used here is based on a 2nd order Taylor approx around the default value, in order to minimise the number of simulations required.*

Apologies: the original manuscript mis-spoke in calling the emulator a Taylor series. In fact, the emulator is a polynomial interpolation:

"As input to the minimization routine, we must provide $\partial m_i/\partial p_j$ and $\partial^2 m_i/\partial p_j^2$ for each $i$ and $j$. These are calculated by fitting a parabola in $p_j$ to $m_i(p_j)$. The parabola passes through the 3 points formed by the output of the default simulation and the outputs of the 2 simulations that perturb $p_j$. Therefore, at these three points, QuadTune's parabolic emulator is an exact match to

the global model solutions. Despite the resemblance of the quadratic emulator to a Taylor series, it is, strictly speaking, a polynomial interpolation, rather than a Taylor series. That is, the derivatives — $\partial m_i/\partial p_j$ and $\partial^2 m_i/\partial p_j^2$ — are not guaranteed to match the global model's derivatives at the default parameter value. "

A polynomial interpolation more likely to be valid away from the default (but within the high and low perturbations) than is a (local) Taylor series. So as long as we stay within the 3 simulations, the results should be reasonably close. To give the reader an intuitive sense of when the emulator is suspect, we have introduced a cartoon that depicts how the emulator is fit to the global model runs (new Fig. 1).

*A calibration method is only as good as the emulator underpinning in it, and it's not clear to me that the emulator in the example is good (perhaps because it's being used to extrapolate too far from the default), which will be leading to biases in the 'optimal' values, and hence incorrect conclusions about what is structural error, and subsequent analysis/conclusions based around this – it is unclear whether a lot of the results and comments in Section 7 would still be true if the emulator were more accurate:*

Section 8 now quantifies the sensitivity to some different choices of emulator shape. It is impossible to prove in any realistic case that the structural error predominates over parametric error, even with a perfect emulator. However, in our example, the RMSE values suggest that structural error predominates.

(As a side comment, we note that even simpler emulators are being published by GMD. For instance, this recently published emulator is linear and doesn't include any 2nd-order terms!: https://gmd.copernicus.org/articles/18/6177/2025/gmd-18-6177-2025.pdf. Presumably GMD is publishing these simple emulators because some users have very expensive (e.g., high-resolution) atmospheric models, don't have access to large computing resources, and hence need to make compromises.)

*There needs to be some validation of the emulator, to check its accuracy. This is partially done, in recognising post-hoc that it predicts it will remove the bias for some regions, but doesn't get close (line 405 'QuadTune's prediction is imperfect'), and in the preceding few lines, where we see it predicts almost zero bias for some locations, but in reality hasn't halved it). Emulator validation should be done before validation rather than discovering this error after doing a new GCM run, as the inadequacies of the emulator will affect the optimisation process, and the results shouldn't be trusted.*

In Section 8, we've done this pre-tuning validation by modifying the emulator in two ways: 1) replacing the quadratic emulator with a piecewise-linear emulator; and 2) adding an interaction

term to the quadratic emulator. Then the consequences of these reasonable changes of emulator shape can be seen in the resulting RMSEs in Table 1.

*The example shown has a poorly tuned default (line 392), but this isn't meant to be the use case as the emulator is only good locally - is there an issue here of extrapolating too far from the default, where the Taylor expansion is not going to be valid? Some validation of the emulator is critical (often out-of-sample predictions or leave-one-outs when the emulator is a GP, something appropriate needs showing here). If the predictions are poor, the emulator needs changing before being used for optimisation.*

Our emulator is a polynomial interpolation, not a Taylor series (sorry for the confusion), and we expect that it will work acceptably well within the high and low parameter perturbation values. In fact, the emulator is exact at those high and low bounds.

This expectation is tested by the piecewise-linear emulator and the interaction-including emulator, which give a sense of the sensitivity to emulator choice.

*Line 551, elsewhere – 'nearby location' in parameter space, how defining nearby? How checking not extrapolating too far? Line 227 mentions can't be large as this violates the Taylor assumption (this sentence would be good in Section 5 when introduce the emulator). What is large?*

"Nearby" is defined as between the high and low perturbed parameter values, which we're assuming have been chosen sensibly by an expert. Line 227 has been revised to:

"if any component of $\delta\boldsymbol{p}$ strays outside the low or high values of the sensitivity runs, then it is more likely there will be a violation of the assumption of QuadTune's emulator that $\mathcal{G}$ can be represented by a simple quadratic interpolation. "

*The problem in the example could be because the optimisation is allowing it to extrapolate too far from the default; it could be because the parameter perturbation used was too large for the Taylor approx to be valid; or some other reason – but whatever the case, there's error in the emulator, and this will affect optimisation of the loss function, and hence optimal parameters and conclusions about patterns of structural error (and what is in fact structural error – this is likely to result in a map that is combination of structural error and parametric uncertainty,*

It is impossible for anyone to prove that parametric error is not contaminating the results, in a realistic problem, but the sensitivity studies in Section 8 suggests that structural error predominates.

*as demonstrated by the fact that the hand-tuned version is 'better' in terms of RMSE).*

We don't put undue significance on the modest difference between the hand-tuned RMSE (10.1 W $m^{-2}$) and the quad-tuned default-configuration RMSE (10.4 W $m^{-2}$). Both are pretty good results. We note that the hand-tuned run has a low RMSE but a large global-mean bias. Also, if quad-tune iterates only one extra time, it produces a RMSE of 10.0 W $m^{-2}$, which (slightly) beats the hand-tuned RMSE. The revised manuscript notes that

"For example, as an initial exploratory experiment, we have started with the iteration-1 optimal parameters from QuadTune (which produce RMSE = 10.4 W $m^{-2}$), constructed a new batch of $2P + 1$ simulations with reduced parameter perturbations about the iteration-1 optimum, re-quad-tuned once, and found an improved RMSE of 10.0 W $m^{-2}$ (not shown)."

A simple quadratic emulator is indeed capable of producing RMSE comparable to hand tuning.

*The emulator seems to be poor for 6_14, 6_18.*

Looking at Fig. 3, one can see that even in 6_14 and 6_18, QuadTune predicts a reduction in bias (Fig. 3b), and in the global simulation, the bias is indeed reduced a lot (Fig. 3c), as these things go. QuadTune has made a useful prediction, even if it is not quantitatively precise.

*Perhaps a way to overcome or explore this issue is to remove or downweight these (and any other) regions where the emulator is poor, and what the suggested optimal is.*

Thanks for the suggestion of removing 6_14 and 6_18, but it is out of scope for this manuscript. This manuscript focuses on the strongest structural errors, which lie in 6_14 and 6_18.

*The model is 'badly out-of-tune', and some bias is removed by QuadTune 'despite the simplicity of the emulator'. The model being badly out-of-tune suggests it's relatively trivial to find some better parameter estimates, and given the identified issues that the emulator has at making predictions in at least some regions, is this 'despite the inaccuracies' of the emulator? I.e., it's found better parameters, despite being inaccurate when extrapolating, because it was straight forward to do so? It might be better to also show a use-case where the default is already much better: in this case, the choice of emulator should be more valid and accurate as only looking locally (as was the initial assumption), and any improvement will be less down to chance.*

An important use case of QuadTune is tuning models that are badly out of tune because, e.g., a parameterization has been modified. In this case, the model is under development, and we want a quick, approximate retuning, before the model structure changes. This is a use case that is not being served by the standard, expensive PPE methods in common use.

In our use case, an approximate emulator is still useful. Yes, a better emulator is preferable if

computation cost is not a consideration, but it is hard to improve the emulator without requiring more (expensive) global runs, which is a serious drawback. Some groups may have access to large computers, but others don't. QuadTune is targeted at the latter.

*The inaccuracy for 6_14 and 6_18 might be hiding/drowning out biases that \*can\* be tuned out (but that don't lead to as large reductions as the incorrect predictions for these regions). E.g. line 434 'QuadTune strives to reduce the bias in 6_14, 6_18 at the expense of other regions', but it only does so because the emulator predicts this bias to be (incorrectly) reduced by much more than it actually can be. So these conclusions and discussion are perhaps only relevant conditional on the poor emulator. It's also 'prioritising bias reduction' in these regions because of the choice of weightings.*

If the goal is to reduce bias over the globe rather than do an experiment to learn more about a specific regional bias, then any reasonable tuning method would strive to reduce errors in 6_14 and 6_18 because those errors are largest and hence contribute the most to the loss function.

Our weighting is a simple areal weight. Any similar weighting is inevitably going to prioritize 6_14 and 6_18 because those biases are large.

Any modeling center that wants to release an operational model would want to explore reducing those errors. Luckily, QuadTune's recommended parameter values do lead to a substantial improvement in those regions, even if QuadTune's prediction is over-optimistic.

Yes, exploring other regional biases can be done by up-weighting those regions, but that is beyond the scope of this manuscript.

*With a more accurate emulator, parametric uncertainty might be reduced much more, and in a different way, trading off other biases – e.g. Figure 4 shows that getting increased biases at some locations as trade-offs for improvements that don't actually exist, which is problematic. The fact that the 'hand-tuned' version has found a better set of parameters (which itself are likely not optimal) show that QuadTune has not removed parametric uncertainty.*

The sensitivity to choice of emulator is estimated in Section 8.

Looking at Fig. 3b and Fig. 3c, one can see that QuadTune's prediction of the bias is qualitatively similar to the actual bias that results from running a global EAM simulation. Even in 6_14 and 6_18, QuadTune predicts a reduction in bias, and in the global simulation, the bias is indeed reduced a lot, as these things go. QuadTune has made a useful prediction for a model developer.

(Note also that the hand-tuned simulation, shown in Fig. 2b, has a worse global-mean bias than does the quadtuned simulation, shown in Fig. 2c.)

*Figure 2 could demonstrate the accuracy of emulator in a clearer way – e.g., plot QuadTune predictions (b) vs true values (c), or predicted reduction vs actual reduction. Is it just poor for the 2 mentioned regions, or systematically wrong elsewhere?*

We prefer the map because it gives a sense of the spatial pattern of the errors. One can see, e.g., that the broad spatial patterns are reproduced by QuadTune, even if there are errors in individual boxes.

*2. For identifying structural error, need to ensure have removed parametric error. The purpose of tuning is described in the abstract as removing parametric error and leaving structural error behind, and line 37 talks about needing 'guidance on what structural errors remain after parametric errors tuned out'. Slightly caveated by 'gives hints about nature of structural error', but this can only be done with confidence if have removed parametric error. Throughout, the two sources of error are combined (eq 3, 27), and results are a sum of the two. The example given suggests that all parametric error has not been removed, and so the observed biases are some mix of the 2.*

It would be useful indeed to have a guarantee that the global optimum has been found, but no such guarantee is possible in a realistic case. There is always the possibility that a better solution exists for a parameter set that has not been evaluated.

*Usually in calibration exercises using best-input based methods such as Bayesian calibration and History matching, parametric, observational, and structural uncertainties are treated separately. Better justification of why it's fine to combine these here (and ignore observational error) would be helpful.*

By implementing alternative emulator shapes (piecewise-linear and interaction-including), we treat the emulator error separately.

*Line 81 says the model doesn't match the observations for any p, and e.g. eq (3) includes this error term. But line 139 assumes that 'near the optimal values, the model output is an approximate match' and ignores the error. Is this a valid assumption?*

The manuscript now includes the clarifying sentence: "This assumption is valid if the model structural error and observational error are not too large."

Section 3 is intended to be a didactic, toy example that gently introduces the reader to the math and "physics" behind the regression problem that we're doing. The gentleness of the introduction would be impaired by too much mathematical rigor early on.

*Other emulators used in these types of methods use things such as Gaussian processes, to quantify*

*the parametric uncertainty and have some understanding of how accurate the emulator is. Here, ignoring this error and treating as a perfect model (equivalently, assuming constant error at all p). This might be valid very locally around the default, but not generally. Further justification of the choice of emulator, and how 'local'/'nearby' it is valid, is needed, as in the example this is clearly not true (perturbing by too much?). Is the example a good illustration, or is it being used in a way it shouldn't be?*

The emulator error is now explored by two sensitivity studies.

The emulator is expected to be valid between the high and low perturbed parameter values.

The tuning example arose in the course of our model development. It is a real case.

*3. Explanation of the steps in QuadTune in Section 3 could be clearer, and possibly a general re-ordering would be helpful for this. Currently the outline of QuadTune is given in Section 3, but relies on things not yet mentioned, in particular the main descriptions are not given until Section 5. It might be better to first define the key elements, then give overall algorithm, then demonstrate, e.g. Section 2, then 5, then 3, then 4, 6 etc. More specifically:*

How a manuscript is written ought to depend on how it is likely to be read. We assume that most readers will not read the article straight through from front to back. In fact, many readers will not read the entire manuscript. Instead, a typical reader will quickly harvest the manuscript for information. Hence, we have adopted a "newspaper" style, in which the most important information is put up front, and the details are deferred until later. In my verbal discussions, I have found that explaining the outline of QuadTune (Section 3) early is helpful in order to orient the listener, even if not all details can be described at first. Therefore, we prefer to keep the order as is.

*'linearly added to the loss function' in step 2 – up to here there's no mention of a 'loss function' except in the description of the sections in the intro, and I don't think it's addressed properlyl until Section 5.*

To avoid mention of the loss function, the sentence has been reworded to "For illustration, this paper tunes a single field, SWCF, but QuadTune allows multiple observables, e.g., SWCF and surface precipitation, to be tuned simultaneously with user-specified weighting on each observable."

*Similarly, the description of step 3b made me question why the simulations are being designed in this way, and this only become clear in Section 5 when the emulator was explained.*

In the original manuscript, it was already clear in step 3b: "This one-at-a-time sampling strategy determines the quadratic emulator with the minimum number of global simulations."

*Some of the comments within the steps of the algorithm aren't really required in line as they're not parts of the general method, and might be better discussed after, so that the steps of QuadTune are concisely and clearly communicated. E.g., 'for illustration, this paper tunes a single field' is not required for Number 2 of QuadTune.*

The mention of the fact that multiple fields can be tuned simultaneously is helpful for defining what $N$ (the number of regional metrics) means.

*Step 6 of QuadTune is given as 're-run', but this contradicts its mention as a possible extension of QuadTune in Section 9 – is this already part or not?*

Step 6 describes a merely optional, qualitative exploration of tuning trade-offs, whereas Section 9 describes a systematic approach to improve parameter values. To clarify, Step 6 now says *"Optional: If desired, re-run QuadTune in order to explore tuning trade-offs. Such experiments might delete ineffective parameters or more heavily weight a regional metric in order to see how it influences the optimal parameter values."*

*In step 5, 'run QuadTune' – it is not explained at this point what this actually means, which makes me think this overall algorithm should come much later after 'QuadTune' has been developed.*

To clarify what 'run QuadTune' means, we have broken the steps into two parts: Preprocessing, and Tuning/Analysis. This makes it clearer that 'run QuadTune' refers only to the tuning step after the data prep work has been done.

*Line 116 mentions 'the quadratic emulator' but this has not really been mentioned yet. Similarly line 133. 'QuadTune's emulator' is explained in Section 5, but in the QuadTune algorithm there should be some clearer explanation of this – could be as simple as 'emulate model output, use this to tune parameters/minimise loss function' – I didn't feel this was spelt out until much later, and overall clarity would be aided by this.*

The original manuscript introduced the emulator in the introduction: *"QuadTune carves the globe into regions and approximates the model parameter dependence by use of an uncorrelated quadratic emulator."*

*Step 3 is the first mention of 'default' – are these standard, or another user choice?*

"Default" would often be the standard setting but it could also be a non-standard user choice. That is common usage of the word "default."

*4. The main example would benefit from slightly more explanation at the start – some of the as-*

*sumptions being used here are spread throughout other parts of the paper, whereas it would be clearer to describe the assumptions you're making in order to run QuadTune in this particular case at the start of Section 7. E.g.,*

*Fig 3 – mentions the tunable parameters are defined in B1. Say this at the start of the example section, make clear what the parameters are, what P is, etc.*

Good idea. Section 7 now includes the sentences "We tune $P = 5$ tunable parameters with the names c8, n2_thresh, sfc, n2, and n2_wp2. They are defined in Table B1."

*What are the perturbations? I don't think these are mentioned. Stating the default values and the range they vary in in Table B2 might be helpful.*

In the original manuscript, the perturbations were displayed in the three-dot plot (Fig. 4), but in the revised version, we have also listed the numerical values in Table B1.

*Could be more specific about the hand-tuned version – what are the optimal parameters chosen here, how compare to QuadTune optimal? Are they at least in a similar region of parameter space or moved in different directions from the default? Do the bias patterns (when aggregated to regions, like in Fig 2) look the same?*

Yes, Fig. 2 shows that the bias pattern from the hand-tuned run (Fig. 2b) is similar to the quadtuned run (Fig. 2c).

*Can you be more specific about the number of simulations done for hand-tuning than 'dozens' (was it done specifically for this comparison or existed already?) These other simulations exist and so could possibly be used as out-of-sample points for assessing accuracy of the emulator.*

Unfortunately, the hand tuning took place over many months before this paper was conceived, and we no longer have a record of the exact number of hand-tuning runs we did. Most of the hand tuning runs have not been saved.

*5. I think consideration of other sensitivities is important to demonstrate the method works. Section 8 considers sensitivity to size of regions and duration, but I think there would be larger sensitivities to other assumptions in the method:*
*Weighting*
*Choice of emulator*
*Size of parameter perturbation*
*Choice of default*

Section 8 is intended to explore the sensitivity of QuadTune's RMSE to the configuration of Quad-Tune itself (e.g., size of regions, duration of global simulations), rather than the sensitivity of the results to what question the user is asking.

The revised Section 8 now presents two sensitivity studies on the choice of emulator, which is a configuration aspect of QuadTune itself.

We also showed in

The other choices (weighting, default parameter values) can and should be explored by the user, depending on what aspects of the model behavior matter to him. They are not QuadTune configuration choices of the kind that we wish to explore in Section 8.

*The example presented does worse than hand-tuning, and the emulator is extrapolating inaccurately. Does changing emulator/perturbations make this more accurate and out-perform hand-tuning? Does weighting differently remove different patterns of bias and hence lead to different conclusions about structural error? This seems important as the purpose of the method is to give insight about this. Even in the context of the comparisons already made, with little different in the RMSE, do these lead to very different parameter estimates / pattern of bias? I think there's work to convince that the method can lead to reasonably robust results, and is not being influenced by an inaccurate emulator or other strict assumptions (re. structure of errors).*

The revised version of Section 8 shows that the addition of one interaction term in the emulator function allows QuadTune to equal the performance of hand tuning.

Section 9 now notes that iterating one extra time allows QuadTune to improve the hand-tuned RMSE.

Of course changing the weighting will change the tuning trade-offs. E.g., if a user de-weights a biased region, then QuadTune will no longer need to prioritize that region at the expense of other regions. Even with a perfect emulator, any tuner will give different optimal parameter values given different weights. Choosing different weights changes the question asked. Although it is a good way to understand different aspects of the behavior of the global model, it is beyond the scope of this manuscript.

*The paper would also benefit from an example where the emulator + optimisation are demonstrated to work well, as this combination is 'QuadTune'. Currently the optimal results, and hence downstream explanations, are being driven by the emulator predicting it removes biases where it can't. Perhaps the better use case is when the default is already reasonably good, so the choice of simulation design*

*and Taylor approximation is valid as truly looking locally only, but I'm not sure.*

The example in the manuscript is an example where QuadTune works well. It reduces RMSE of SWCF from 12.4 to 10.4 W m$^{-2}$ with only 11 global simulations! We expect that QuadTune will prove to be a useful tool for model developers. It has already been useful to us.

*6. Notation – there's a few places where there's inconsistencies or undefined terms, including:*

*In Section 2, line 90 N refers to the number of regions. In Section 3, line 109 it's this multiplied by number of fields. Easier to follow if don't re-define.*

Now Section 2 uses $n$ for the number of regions.

*Line 91 – a region is defined as x. In Section 4 they're defined a bit differently, with e.g. x in Sc, ideally do so consistently.*

The notation on Line 91 has been made more consistent with that in Section 4.

*Eq 10 – bold p, b not been explicitly defined. Similarly, the other vectors in (11) and (12) not explicitly defined.*

In the original manuscript, the symbols in Eqn. (10) were defined by giving a matrix form of the equation in Eqn. (9) and then stating:

" . . . rewritten in symbolic form,

$$\mathbf{S} \cdot \delta \boldsymbol{p} \approx -\delta \boldsymbol{b}. \tag{1}$$

We regard as knowns the bias vector $\delta \boldsymbol{b}$ and the sensitivity matrix $\mathbf{S}$."

Likewise, the symbols in Eqns. (11) and (12) were defined in the original manuscript by the following text:

Now suppose that we use least-squares linear regression to find optimal parameter perturbations $\delta p_{opt,j}$. Then let us define

$$\delta b_{remov,i} \equiv -S_{ij} \delta p_{opt,j}. \tag{2}$$

Here, $\delta \boldsymbol{b}_{remov}$ is the default model output minus the tuned model output. The vector $\delta \boldsymbol{b}_{remov}$ is the part of the bias $\delta \boldsymbol{b}$ that is removable by linear regression. Thinking more geometrically, $\delta \boldsymbol{b}_{remov}$ is the part of the bias vector $\delta \boldsymbol{b}$ that lies within the subspace spanned by the columns of $\mathbf{S}$.

Now define the residual bias, $\delta \boldsymbol{b}_{resid}$, as the part of the bias that remains after $\delta \boldsymbol{b}_{remov}$ has been removed by linear regression:

$$\delta \boldsymbol{b} \equiv \delta \boldsymbol{b}_{remov} + \delta \boldsymbol{b}_{resid}. \tag{3}$$

Here, $\delta \boldsymbol{b}_{resid}$ is the tuned model output minus the observational values. (Note that the residual bias $\delta \boldsymbol{b}_{resid}$ is defined to have the opposite sign as the residual that is traditionally defined in statistics.)"

The vectors were defined in the original manuscript.

*Line 140 – the equations here are assuming p_1, p_2 are the 'optimal' values, but should these be distinguished by e.g. p_1\*, p_2\* (see e.g. best input approaches, optimal commonly written x\* or theta\*).*

Thanks, that is clearer.

*Section 5.1 – defines regional metrics for the first time as m_i. The same type of regional metrics were written in a different way in Section 4, and for consistency should probably use the m_i style notation there as well.*

Section 4 discusses a simple toy example, and for that purpose, we prefer a less abstract notation than $m_i$.

*Eq 19 – m_obs isn't defined*

It is now defined.

*Eq 21 – j, k not defined. Compared to Eq 22, this equation is missing sums?*

We've now included the sums, which effectively defines the indices $j$ and $k$.

*Eq 26, and several places thereafter – (no sum over i) – no need to say this, self-evident from the equations. Could write i = 1, ... N to be clear doing for each metric.*

Stating "no sum over i" is a standard way of clarifying that we're not using Einstein summation convention.

*Other comments:*

*Line 402 – phrases like 'QuadTune thinks' are used in several places. I'd much prefer phrasings like*

*'QuadTune predicts', as that more accurately describes what's actually happening.*

The meaning of "QuadTune thinks" is clear.

*Line 122 - 'A possibly weighted version of Eq (5)' – I think everywhere it is assumed that this is weighted geographically, so Eq 5 should probably include weights. Can comment that the equal weighting case is then a special case if that, if it's actually useful.*

We prefer not to introduce the weights as early as Equation 5. The weights are a technical aspect. In Equation 5, we're just introducing the regional tuning problem, not discussing the details of QuadTune.

*Line 153 – 'similarly for delta p_2' – could just define in terms of p_i, so that then works for general case later as well.*

This is a toy example for illustrative purposes. I think that it will be simpler for non-mathematical readers to speak here of concrete parameters, $p_1$ and $p_2$, rather than the more abstract $p_i$.

*The phrasing 'stubborn bias' is defined/explained on page 10, but has already been used on pages 1, 5 and 9. I don't think it's obvious how this is being defined until p10, so perhaps just be explained earlier.*

Giving a technical definition of "stubborn bias" earlier would disrupt the flow of the discussion. The phrase "stubborn bias" is intuitively evocative, and so we prefer to leave the definition later.

*Line 255 – because of the ordering, with Section 5 after the algorithm is given, 'requiring an extra P global simulations' could be read as on top of the 2P+1 simulations that were mentioned earlier, rather than these already being part of it.*

To clarify, the sentence has been revised to: "Including these extra terms has the drawback of requiring an extra $P$ global simulations beyond the $P+1$ simulations needed for a linear emulator, leading to a total of $2P+1$ simulations."

*Line 275 – 'simplicity of the approximation helps us better understand structural errors' – this is only true if the approximation is accurate, which the example suggests it is not (perhaps because in this case 'nearby' is defined too widely). Need some mention about sizes of perturbations, checking accuracy of local approximation?*

In any tuning problem, one can never be sure that all parametric error has been removed; it is always possible that a better parameter set lies in an unexplored region of parameter space. In our example,

it is true that not all parametric error has been removed, but this is to be expected in any affordable tuning experiment. We believe that the approximation in our example is accurate enough to be useful. The RMSE goes from 12.4 to a respectable 10.4, which is a large improvement for SWCF.

The original text already included mention of the local nature of the approximation ("This local approximation prevents QuadTune from doing more than seeking a nearby local minimum in parameter space") and this is not the spot in the manuscript to add more. However, in the revised Section 8, we have included a sensitivity study that changes the emulator to a piecewise linear one, thereby providing a sense of the emulator error.

*Fig 3 – each row should have a common y axis range.*

A common y-axis would make less apparent the details of the individual panels. We prefer to leave the figure as is. The red, horizontal line that shows observations helps to indicate which tuning results underestimate the obs vs. those that overestimate it.

*Line 513 – 'region 3_6 cannot be improved by tuning' is quite a strong statement. It might be if you varied parameters jointly, fit a different emulator? I.e. it's conditional on your assumptions that you can't improve it.*

One can never prove, in a realistic problem, that there is no corner of parameter space where an improved answer lies. To avoid confusion, we have reworded to state:

"Because each parabola curves away from the observed value of SWCF, any large parameter perturbation worsens the fit. This worsening is 'internal' to region 3_6, rather than a tuning trade-off with other regions."

*Line 515 – 'if wish to remove a residual bias...remains after tuning' – I don't think you have to find another metric or change your model. Within this method, you could also change the weighting of that region? You could also improve the emulator.*

This is intended as advice for a QuadTune user rather than a QuadTune developer. So improving the emulator is not considered an option here. But yes, a user could easily change the weighting. The sentence has been revised to state: "If a QuadTune user wishes to remove a residual bias in a particular regional metric that remains after tuning, then he must either find another global-model parameter to tune, or else make a model structural change, or else upweight the region."

*1st paragraph of Section 7 is a description of QuadTune's properties, rather than part of the example – better in the QuadTune section?*

The example is intended mostly to "illustrate the use of these diagnostic plots". And so it makes sense to have an introductory paragraph in Section 7 that summarizes the intention of these plots.

*Section 8 is extremely short, can probably include as a subsection if remains this short.*

We've added more sensitivity studies to this section, and so it has more content than it did in the original submission.

We prefer to keep Section 8 as a separate section, because the sensitivity of the tunings to configuration changes is a topic that many readers will want to see. We want to make this topic easily discoverable by a reader perusing the article.

**Minor edits/typos:**

*Line 8 – 'through the use of'?*

Changed as requested.

*Line 9 – 'explainability' of what?*

The manuscript now writes "explainability of tuner behavior".

*Line 21 – 'improvement in the overall...'*

Whoops, thanks.

*Line 26 – 'Big gains... often come from structural model improvements' should probably be referenced*

The manuscript now references Vitart (2014).

*Line 50-51 – quite similar to lines 46-48, probably combine these 2 paragraphs and be less repetitive.*

We'd prefer to keep both paragraphs. The first paragraph tells us what information QuadTune provides, and the second paragraph states that this information is visualized in plots. We'd like to keep the second paragraph distinct in order to emphasize the value of our plots. We have added a little detail to the second paragraph: "These plots indicate the manner in which QuadTune has reduced regional biases and the character of the structural errors that prevent further bias reduction."

*Line 77 – 'particular' said twice, only needs one.*

We've kept "particular" twice because the first time it refers to an output and the second time it

refers to a choice of parameter values.

*Line 80 – does f_obs need x dependence here? Often does in what follows.*

We've now included x dependence.

*Line 88 – 'at fine resolution' – I'm not sure you've defined x as being 'fine' resolution, just lon/lat. Maybe say something like across all lon/lat instead.*

The distinction we want to make is between an integral and a sum. The revised manuscript now states "One could attempt to minimize the error at fine resolution with an integral over the whole globe"

Equation 4 should look OK now.

Changed as requested.

*Line 166 – 'we regard as knowns...' would sound better as 'we regard the...as knowns'*

We like having "We regard" and "as knowns" together, rather than separated by the long phrase "the bias vector $\delta\boldsymbol{b}$ and the sensitivity matrix $\mathbf{S}$".

The manuscript now says "The $j$th column of $\mathbf{S}$ tells us how perturbing the $j$th parameter, $\delta p_j$, affects the spatial pattern"

Fixed.

The manuscript now writes "Even when parameter perturbations extend to the limits recommended by expert judgment"

Thanks.

Thanks.

We hear "off of" more often, and so we'd prefer to go with that.

*Line 393 – 'much of the bias' – perhaps too optimistic – only reduced by $\widetilde{2}0\%$*

For these sort of global models, a RMSE of SWCF of 10.4 W m$^{-2}$ is good. It requires a lot of effort to reduce the RMSE by significantly more.

Good catch.

*Line 459 – 'the first singular vector' of what? (Mentioned in Figure caption, but not clear from text).*

The manuscript now writes "The first singular vector of the quasi-linear sensitivity matrix $\mathbf{S}^+$"

Whoops, thanks!

---

## Author Response (AR2)

**Responses to 2nd round of reviews of "QuadTune version 1: A regional tuner for global atmospheric models"**

**Vincent E. Larson, Zhun Guo, Benjamin A. Stephens, Colin Zarzycki, Gerhard Dikta, Yun Qian, Shaocheng Xie**

We thank the reviewer for re-reading the manuscript and making further comments.

The reviewer comments are repeated below in blue italics, with responses interspersed in a non-italicized font. Changes to the manuscript text appear in orange.

**1    Responses to Reviewer**

*Thanks for taking the time to respond in detail to the comments. I'm mainly happy with the changes made, with some small specific comments below.*

*In particular, I think the description of what QuadTune is (Section 3) is clearer, and the ordering is fine now that this references ahead. Figure 1 gives a nice illustration. The extension of Section 8 to compare more sensitivities is valuable as well, and whilst the difference between 10.1 and 10.4 is perhaps not large, it's nice to see that can match the hand-tuned version even more closely with few simulations and a slightly more complex emulator, based around parameter interactions that are most important. I think this is a useful additional demonstration and better shows that the tool is working well.*

Thank you.

*"As a side comment, we note that even simpler emulators are being published by GMD. For instance, this recently published emulator is linear and doesn't include any 2nd-order terms! (link removed). Presumably GMD is publishing these simple emulators because some users have very expensive (e.g., high-resolution) atmospheric models, don't have access to large computing resources, and hence need to make compromises."*

*Is it worth referencing this paper as motivation that this is a reasonable approach? E.g., around line 160 mention GPs and PC, but could perhaps reference the fact that simpler approaches are also used — for the reasons you mention here.*

This reference is now cited in the manuscript with the following comment:

"However, other authors have gone further and dropped the entire quadratic term, including the diagonal part, in order to reduce the cost (e.g., Petrov et al., 2025)."

*"Stating 'no sum over i' is a standard way of clarifying that we're not using Einstein summation convention."*

*I'm not convinced this is necessary as it's already clear from the summation only being over j (eq 26), or lack of summation notation (31). If using this clarification, for consistency does it then need adding to things like (25)?*

(25) does not need to state 'no sum over i' because in (25), the $i$ is not repeated within a term, and hence the summation convention is not violated. There is just one $i$ in each term. However, in other equations in the revised manuscript, we've added 'no sum over i' in order to clarify.

*Line 151 – it's clear from context, but maybe explicitly say something like 'near the optimal parameter values (p1,opt, p2,opt)' (like with definition of default values in line 162)*

Thanks. Changed as suggested.

*Line 549 – 'they' instead of 'he'?*

If the manuscript were to use 'they', then the pronoun wouldn't agree in number with the antecedent noun, causing unclarity.

*Line 549 – could delete the first 'or else' in this sentence?*

We prefer to keep both 'or else' phrases because it adds clarity.

*Table 1 – some are to 1DP, some 2DPs – is 10.3 really 10.30, or 10.25?*

We have modified the table and text such that only one decimal place is retained.